# DICE: End-to-end Deformation Capture of Hand-Face Interactions from a Single Image

**Qingxuan Wu**[1], **Zhiyang Dou**[1,2,7,*], **Sirui Xu**[3], **Soshi Shimada**[4], **Chen Wang**[1], **Zhengming Yu**[6]
**Yuan Liu**[2], **Cheng Lin**[2], **Zeyu Cao**[5], **Taku Komura**[2,7], **Vladislav Golyanik**[4]
**Christian Theobalt**[4], **Wenping Wang**[6], **Lingjie Liu**[1,*]
[1]University of Pennsylvania, [2]The University of Hong Kong
[3]University of Illinois Urbana-Champaign, [4]Max Planck Institute for Informatics
[5]University of Cambridge, [6]Texas A&M University, [7]TransGP

## Abstract

Reconstructing 3D hand-face interactions with deformations from a single image is a challenging yet crucial task with broad applications in AR, VR, and gaming. The challenges stem from self-occlusions during single-view hand-face interactions, diverse spatial relationships between hands and face, complex deformations, and the ambiguity of the single-view setting. The previous state-of-the-art, Decaf, employs a global fitting optimization guided by contact and deformation estimation networks trained on studio-collected data with 3D annotations. However, Decaf suffers from a time-consuming optimization process and limited generalization capability due to its reliance on 3D annotations of hand-face interaction data. To address these issues, we present *DICE*, the first end-to-end method for Deformation-aware hand-face Interaction reCovEry from a single image. DICE estimates the poses of hands and faces, contacts, and deformations simultaneously using a Transformer-based architecture. It features disentangling the regression of local deformation fields and global mesh vertex locations into two network branches, enhancing deformation and contact estimation for precise and robust hand-face mesh recovery. To improve generalizability, we propose a weakly-supervised training approach that augments the training set using in-the-wild images *without* 3D ground-truth annotations, employing the depths of 2D keypoints estimated by off-the-shelf models and adversarial priors of poses for supervision. Our experiments demonstrate that DICE achieves state-of-the-art performance on a standard benchmark and in-the-wild data in terms of accuracy and physical plausibility. Additionally, our method operates at an interactive rate (20 fps) on an Nvidia 4090 GPU, whereas Decaf requires more than 15 seconds for a single image. The code will be available at: `https://github.com/Qingxuan-Wu/DICE`.

## 1 Introduction

Hand-face interaction is a common behavior observed up to 800 times per day across all ages and genders (Spille et al., 2021). Therefore, faithfully recovering hand-face interactions with plausible deformations is an important task given its wide applications in AR/VR (Pumarola et al., 2018; Hu et al., 2017; Wei et al., 2019), character animation (Qin et al., 2023; Zhao et al., 2024), and human behavior analysis (Liu et al., 2022; Guo et al., 2023; Mueller et al., 2019). Given the speed requirement of downstream applications like AR/VR, fast and accurate 3D reconstruction of hand-face interactions is highly desirable. However, several challenges make monocular hand-face deformation and interaction recovery particularly challenging: **1**) self-occlusions involved in hand-face interaction, **2**) the diversity of hand and face poses, contacts, and deformations, and **3**) ambiguity in the single-view setting. Most existing methods (Rempe et al., 2020; Muller et al., 2021) only reconstruct hand (Romero et al., 2022) and face (Li et al., 2017) meshes, unified as a whole-body model (Loper et al., 2023; Pavlakos et al., 2019), without capturing contacts and deformations. A seminal advance, Decaf (Shimada et al., 2023), recovers hand-face interactions while accounting for both deformations and contacts. However, Decaf requires time-consuming optimization, which takes more than 15 seconds per image, rendering it unsuitable for interactive applications. Moreover, Decaf's iterative fitting process depends heavily on accurate initial estimates of hand and face keypoints, as well as contact

---

*Corresponding Authors

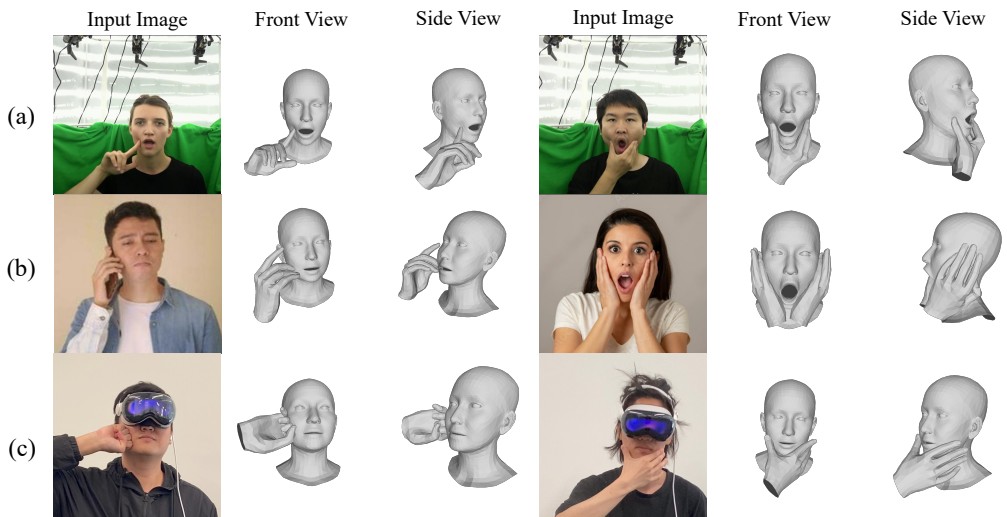

| Input Image | Front View | Side View | Input Image | Front View | Side View |

Figure 1: Our method is the first end-to-end approach that captures hand-face interaction and deformation from a monocular image. Results are from (a) Decaf's validation dataset, (b) in-the-wild images, and (c) VR use cases.

points on their surfaces, which could fail when significant occlusion is present in the image (See Fig. 8). Additionally, Decaf cannot scale up their training to fruitful hand-face interaction data in the wild, as they require 3D ground-truth annotations, such as contact labels and deformations that are not available from the in-the-wild data.

To tackle the issues above, we present *DICE*, the first end-to-end approach for Deformation-aware hand-face Interaction reCovEry from a monocular image. Our approach features three key designs: **1**) Our Transformer-based model leverages the attention mechanism to capture the relationships between the hand and face. **2**) Motivated by the global nature of pose and shape, as well as the local nature of the deformation field and contact probabilities–their invariance to global transformations of the hand and face–we propose disentangling the regression of global geometry and local interaction into two network branches. We evaluate this approach to enhance the estimation of deformations and contacts while ensuring accurate and robust recovery of hand and face meshes. **3**) Instead of directly regressing the hand and face parameters, we learn an intermediate non-parametric mesh representation. This representation is used to regress the pose and shape parameters of the hand and face using a neural inverse-kinematics network. Compared to directly regressing pose and shape parameters, which learns abstract parameters in a highly non-linear space, predicting vertex positions in Euclidean space and then applying inverse-kinematics improves the reconstruction accuracy (Li et al., 2021; 2023c;b). Combining all these contributions, our model achieves higher reconstruction accuracy than all previous regression- (Feng et al., 2021a; Li et al., 2017; Lin et al., 2021a) and optimization-based (Shimada et al., 2023; Lugaresi et al., 2019; Li et al., 2017) methods. Additionally, by utilizing the neural inverse-kinematics network, our approach benefits from an animatable parametric representation of the hand and face, which can be readily utilized in downstream applications.

Despite containing rich 3D annotations, the existing benchmark dataset (Shimada et al., 2023) collected in a studio is still limited in the diversity of hand motions, facial expressions, and appearances. Training only on such a dataset limits the model's ability to generalize to in-the-wild scenarios. To achieve robust and generalizable hand-face interaction and deformation recovery, we introduce a weak-supervision training pipeline that utilizes in-the-wild images without the reliance on 3D annotations. To achieve this, our key insight is to leverage additional prior knowledge, such as depth supervision alongside 2D keypoint supervision, compensating for the absence of ground truth contact and deformation annotations. We leverage the robust depth prior provided by a diffusion-based monocular depth estimation model (Ke et al., 2024), which provides essential geometric information for accurate mesh recovery and captures spatial relationships critical for contact state and deformation estimation. As the task becomes highly ill-posed for in-the-wild images, we further employ pose priors of the hand and face by introducing hand and face parameter discriminators that learn rich hand and face motion priors from additional datasets on hand or face separately (Pan et al., 2023a; Zimmermann et al., 2019). By incorporating a small set of real-world images alongside the Decaf dataset and leveraging our weak-supervision pipeline, we markedly enhance the accuracy and generalization capacity of our model.

As a result, our method achieves superior performance in terms of accuracy, physical plausibility, inference speed, and generalizability. It surpasses all previous methods in accuracy on both standard

benchmarks and challenging in-the-wild images. Fig. 1 visualizes some results of our method. We conduct extensive experiments to validate our method. In summary, our contribution is three-fold:

- We propose DICE, the first end-to-end learning-based approach that accurately recovers hand-face interactions and deformations from a single image.
- We propose a novel weak-supervised training scheme with depth supervision on keypoints to augment the Decaf data distribution with a diverse real-world data distribution, significantly improving the generalization ability.
- DICE achieves superior reconstruction quality compared to baseline methods while running at an interactive rate (20fps).

## 2   RELATED WORK

Extensive efforts have been made to recover meshes from monocular images, including human bodies (Bogo et al., 2016; Moon & Lee, 2020; Li et al., 2021; Cai et al., 2024; Contributors; Xie et al., 2022; Wang & Daniilidis, 2023; Wang et al., 2023b; Lin et al., 2021b; Kanazawa et al., 2018; Cai et al., 2022; Zhang et al., 2021b; Feng et al., 2023; Li et al., 2022c; Wang et al., 2023a; Dou et al., 2023b; Cho et al., 2022; Huang et al., 2022b; Lin et al., 2021a), hands (Rong et al., 2021; Moon et al., 2020; 2024; Moon, 2023; Oh et al., 2023; Park et al., 2022; Yang et al., 2021; 2022b; Li et al., 2023d; Yu et al., 2023), and faces (Feng et al., 2021b; 2018; Wood et al., 2022; Daněček et al., 2022; Zielonka et al., 2022; Chai et al., 2023; Zhang et al., 2023c; Otto et al., 2023; He et al., 2023; Chatziagapi & Samaras, 2023; Kumar et al., 2023; Li et al., 2023a). This also includes recovering the surrounding environments (Clever et al., 2022; Huang et al., 2022a; Hassan et al., 2019; 2021; Zhang et al., 2020b; Li et al., 2022b; Zhang et al., 2021c; Shimada et al., 2022; Luo et al., 2022; Weng & Yeung, 2021) and interacting objects (Yang et al., 2022a; Zhang et al., 2020a; Pham et al., 2017; Tsoli & Argyros, 2018; Hampali et al., 2020; Tekin et al., 2019; Zhang et al., 2020a; Grady et al., 2021; Pokhariya et al., 2023; Hasson et al., 2019; Ye et al., 2022; Chen et al., 2023; 2021; Liu et al., 2021; Corona et al., 2020) while reconstructing the mesh. The acquired versatile behaviors play a crucial role in various applications, including motion generation (Tevet et al., 2022; Peng et al., 2022; Pan et al., 2023b; Guo et al., 2022; Wang et al., 2022a; Xu et al., 2023; 2024; Lin et al., 2024; Zhou et al., 2023; Wan et al., 2023a; Peng et al., 2021; Dou et al., 2023a; Wan et al., 2023b), augmented reality (AR), virtual reality (VR), and human behavior analysis (Zhang et al., 2023a; Yang et al., 2024; Zhang et al., 2024; 2023b; Guo et al., 2023; Liu et al., 2022). In the following, we mainly review the related works on hand, face and full-body mesh recovery.

**3D Interacting Hands Recovery.** Recent advancements have markedly enhanced the capture and recovery of 3D hand interactions. Early studies have achieved reconstruction of 3D hand-hand interactions utilizing a fitting framework, employing resources such as RGBD sequences (Oikonomidis et al., 2012), hand segmentation maps (Mueller et al., 2019), and dense matching maps (Wang et al., 2020). The introduction of large-scale datasets for interacting hands (Moon et al., 2020; 2024) has motivated the development of regression-based approaches. Notably, these include regressing 3D interacting hand directly from monocular RGB images (Rong et al., 2021; Moon, 2023; Zhang et al., 2021a; Li et al., 2022a; Zuo et al., 2023). Additionally, research has extended to recovering interactions between hands and various objects in the environment, including rigid (Cao et al., 2021; Grady et al., 2021; Liu et al., 2021; Tekin et al., 2019; Fan et al., 2024; Ye et al., 2023b;a), articulated (Fan et al., 2023), and deformable (Tretschk et al., 2023) objects. Following Shimada et al. (2023), our work distinguishes itself by introducing hand interactions with a deformable face, characterized by its non-uniform stiffness—a significant difference from conventional deformable models. This innovation presents unique challenges in accurately modeling interactions.

**3D Human Face Recovery.** Research in human face recovery encompasses both optimization-based (Aldrian & Smith, 2012; Thies et al., 2016) and regression-based (Feng et al., 2018; Sanyal et al., 2019) methodologies. Beyond mere geometry reconstruction, recent approaches have evolved to incorporate training networks with the integration of differentiable renderers (Feng et al., 2021b; Zielonka et al., 2022; Zheng et al., 2022; Wang et al., 2022b; Cho et al., 2022). These methods estimate variables such as lighting, albedo, and normals to generate facial images and compare them with the monocular input. However, a significant limitation in much of the existing literature is the neglect of the face's deformable nature and hand-face interactions. Decaf (Shimada et al., 2023) represents a pivotal development in this area, attempting to model the complex mimicry of musculature and the underlying skull anatomy through optimization techniques. In contrast, our work introduces a regression-based, end-to-end method for efficient problem-solving, setting a new benchmark in the field.

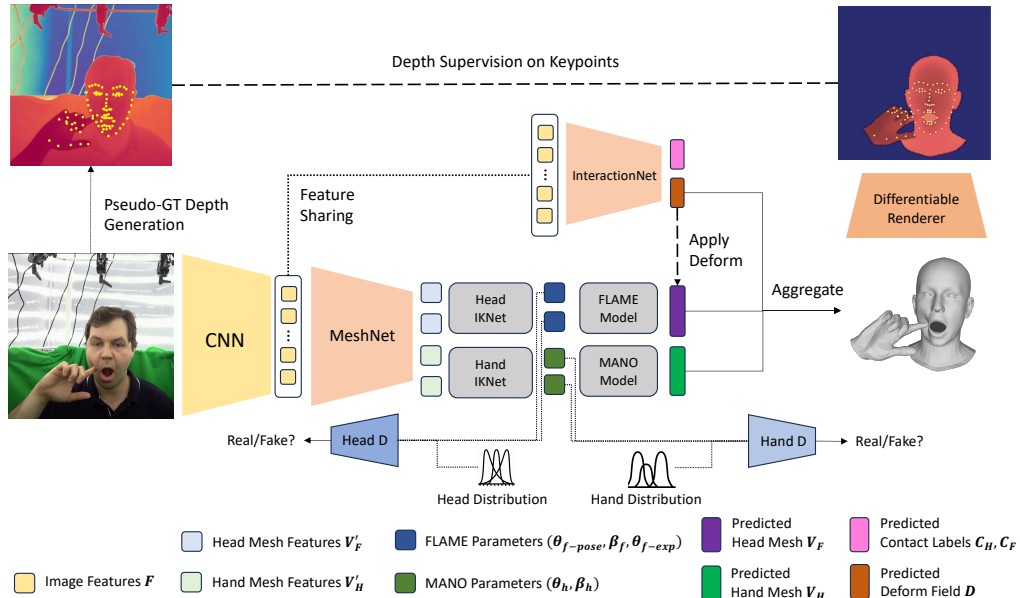

Figure 2: **Overview of the proposed DICE framework**. The input image is first fed to a CNN to extract a feature map, which is then passed to the Transformer-based encoders for mesh and interaction, *i.e.*, MeshNet and InteractionNet. MeshNet extracts hand and face mesh features, which are then used by the Inverse Kinematics models (IKNets) to predict pose and shape parameters that drive FLAME (Li et al., 2017) and MANO (Romero et al., 2022) models. InteractionNet predicts per-vertex hand-face contact probabilities and face deformation fields from the feature map, where the latter is applied to the face mesh output by the FLAME model. To improve the generalization capability, we introduce a weakly-supervised training scheme using off-the-shelf 2D keypoint detection models (Lugaresi et al., 2019; Bulat & Tzimiropoulos, 2017) and depth estimation models (Ke et al., 2024) to provide depth supervision on keypoints. In addition, we use face and hand discriminators to constrain the distribution of parameters regressed by IKNets.

**3D Full-Body Recovery.** The task of monocular human pose and shape estimation involves reconstructing a 3D human body from a single image. Optimization-based approaches (Bogo et al., 2016; Pavlakos et al., 2019; Shi et al., 2023; Rempe et al., 2021) employ the SMPL model (Loper et al., 2023), fitting it to 2D keypoints detected within the image. Conversely, regression-based methods (Li et al., 2021; Lassner et al., 2017; Kocabas et al., 2021; Kanazawa et al., 2018; Feng et al., 2021a; Fang et al., 2021; Lin et al., 2023; Cai et al., 2024; Feng et al., 2023) leverage deep neural networks to directly infer the pose and shape parameters of the SMPL model. Hybrid methods (Kolotouros et al., 2019a) integrate both optimization and regression techniques, enhancing 3D model supervision. Distinct from these approaches, we follow parametric methods (Li et al., 2021; Cai et al., 2024; Kanazawa et al., 2018; Bogo et al., 2016) due to its flexibility for animation purposes. Unlike most research in this domain, which primarily concentrates on the main body with only rough estimations of hands and face, our methodology uniquely accounts for detailed interactions between these components.

## 3 METHOD

**Problem Formulation.** Following Decaf (Shimada et al., 2023), we adopt the FLAME (Li et al., 2017) and MANO (Romero et al., 2022) parametric models for hand and face. Given a single RGB image $\mathbf{I} \in \mathbb{R}^{224 \times 224 \times 3}$, the objective of this task is to reconstruct the vertices of a hand mesh $\mathbf{V}_H \in \mathbb{R}^{778 \times 3}$ and a face mesh $\mathbf{V}_F \in \mathbb{R}^{5023 \times 3}$, along with capturing the face deformation vectors $\mathbf{D} \in \mathbb{R}^{5023 \times 3}$ resulting from hand-face interaction and its non-rigid nature. Additionally, we estimate per-vertex contact probabilities of hand $\mathbf{C}_H \in \mathbb{R}^{778}$ and face $\mathbf{C}_F \in \mathbb{R}^{5023}$.

### 3.1 TRANSFORMER-BASED HAND-FACE INTERACTION RECOVERY

Our model incorporates a two-branch Transformer architecture and integrates inverse-kinematic models, specifically, MeshNet, InteractionNet, and IKNets. A differentiable renderer (Ravi et al., 2020) is used to compute depth maps from the predicted mesh for depth supervision, while the hand and face discriminators are used as priors for constraining the hand and face poses; See Fig. 2 for an overview.

Given a monocular RGB image $\mathbf{I}$, we use a pretrained HRNet-W64 (Sun et al., 2019) backbone to extract a feature map $\boldsymbol{X}_{\mathrm{I}} \in \mathbb{R}^{H \times W \times C}$. Following Lin et al. (2021a;b), we flatten the image feature maps and upsample the $H \times W$ feature maps to $N$ feature maps, corresponding to each keypoint and downsampled vertex of both hand and face. The feature maps $\mathbf{F}' \in \mathbb{R}^{N \times C}$ are then concatenated with the downsampled hand and face vertex and keypoint coordinates of dimension $N \times 3$, with the pose set to the mean pose, serving as positional encodings. This results in the final feature map $\mathbf{F} \in \mathbb{R}^{N \times (C+3)}$. To model the vertex-vertex interactions, we mask the feature maps $\mathbf{F}$ for a randomly selected subset of vertices.

Once the feature map $\mathbf{F}$ is obtained, it is fed into MeshNet and InteractionNet, which handle the regression of mesh vertices and the deformation field separately. This decomposition is motivated by their semantic differences: mesh contains global features, whereas deformation vectors and contact states are localized features, *i.e.*, invariant to the global transformations of the hand and face. Thus, MeshNet takes the feature map $\mathbf{F}$ as input and regresses the unrefined vertex positions of hand $\mathbf{V}'_H$ and face $\mathbf{V}'_F$. InteractionNet, on the other hand, predicts the 3D deformation field $\mathbf{D}$ for each face vertex, along with the contact labels for each hand and face vertex, $\mathbf{C}_H$ and $\mathbf{C}_F$, respectively. Note the contacts and deformations are regressed within the same encoder to model their causal relationship: the contacts cause the deformations. We validate our design in Sec. 4.4.

Next, instead of directly using the unrefined hand and face vertices $\mathbf{V}'_H$ and $\mathbf{V}'_F$, our method takes these vertices as input to regress the pose and shape of their respective parametric models (Li et al., 2017; Romero et al., 2022). This is achieved by a neural inverse kinematics model, named IKNet, following Kolotouros et al. (2019b). The IKNet takes the unrefined hand and face vertices $\mathbf{V}'_H$ and $\mathbf{V}'_F$ as inputs and predicts their pose, shape, and expression parameters $(\boldsymbol{\theta}_{\mathrm{h}}, \boldsymbol{\beta}_{\mathrm{h}})$ for hand, $(\boldsymbol{\theta}_{\mathrm{f\text{-}pose}}, \boldsymbol{\beta}_{\mathrm{f}}, \boldsymbol{\theta}_{\mathrm{f\text{-}exp}})$ for face, along with the root position and orientation for hand $(\boldsymbol{t}_h, \boldsymbol{r}_h)$ and face $(\boldsymbol{t}_f, \boldsymbol{r}_f)$, respectively. Afterward, we use the predicted parameters to first obtain the hand vertices $\mathbf{V}_H$ and undeformed face vertices $\mathbf{V}_F^*$. Then, we apply the deformation $\mathbf{D}$ predicted by the InteractionNet on $\mathbf{V}_F^*$ to get the final deformed face $\mathbf{V}_F$. Utilizing parametric forward-kinematics and neural inverse-kinematics models offer several advantages: first, it enables readily animatable meshes for downstream applications; second, compared to non-parametric regression methods, where meshes typically contain artifacts such as spikes (Lin et al., 2021a; Cho et al., 2022; Lin et al., 2021b), this approach significantly improves mesh quality; third, the compact parameter space allows for a more effective discriminator, which will be discussed in the following section.

### 3.2 WEAKLY-SUPERVISED TRAINING SCHEME

Although the aforementioned benchmark, Decaf (Shimada et al., 2023), accurately captures hand, face, self-contact, and deformations, it consists of only eight subjects and is recorded in a green-screen studio. Thus, training a model only with the Decaf dataset limits its generalization capability to in-the-wild images that exhibit far more complex and diverse human identities, hand poses, and face poses.

To further enhance the generalization capability, we train our model with 500 diverse in-the-wild images of hand-face interaction collected from the internet *without* the reliance on the 3D ground truth annotations. First, we use 2D hand and face keypoints detected by Lugaresi et al. (2019) and Bulat & Tzimiropoulos (2017) as pseudo-ground-truth. Then, we use Marigold (Ke et al., 2024), a diffusion-based monocular depth estimator pre-trained on a large amount of images to generate 2D affine-invariant depth maps for depth supervision (see Eq. 4). The depth supervision provides a strong depth prior, which guides the spatial relationship between hand and face meshes, promoting accurate modeling of hand-face interaction. We first use a differentiable rasterizer (Ravi et al., 2020) to compute a depth map from the predicted hand and face meshes. We use a depth loss to measure the difference between the depths of the hand and face keypoints and their corresponding points on the predicted depth map, providing supervision. This keypoint-to-keypoint correspondence enables accurate depth supervision even when the rendered hand/face mesh and the ground-truth meshes are misaligned. Moreover, we train adversarial priors on the hand and face parameter space on multiple hand and face motion datasets: the face-only RenderMe-360 (Pan et al., 2023a), the hand-only FreiHand (Zimmermann et al., 2019), and Decaf (Shimada et al., 2023). This ensures the plausibility of generated face and hand poses and shapes while allowing for flexible poses and shapes beyond the Decaf data distribution to handle in-the-wild cases. The overall weak-supervision pipeline significantly enhances our model's generalization capability and robustness, which we investigate in Sec. 4.4.

### 3.3 LOSS FUNCTIONS

**Mesh losses $\mathcal{L}_{\mathrm{mesh}}$:** For richly annotated Decaf dataset (Shimada et al., 2023), we employ an $L_1$ loss for 3D keypoints, 3D vertices, and 2D reprojected keypoints, comparing them against their respective ground-truths, following common practice in human and hand mesh recovery (Lin et al., 2021a; Cho et al., 2022; Dou et al., 2023b). We further apply an $L_1$ loss $\mathcal{L}_{\mathrm{params}}$ on the estimated hand and face pose, shape, and facial expression against the ground-truth parameters. For in-the-wild data, only the 2D reprojected keypoints are supervised, as they are the only type with corresponding ground truth.

**Interaction losses $\mathcal{L}_{\mathrm{interaction}}$:** Similar to Shimada et al. (2023), we impose Chamfer Distance losses to promote touch for predicted contact vertices and discourage collision. We also introduce a binary cross-entropy loss to supervise contact labels and a deform loss with adaptive weighting mechanism to supervise deform vectors. For in-the-wild data, we also impose touch and collision losses since they do not require annotations.

**Adversarial loss $\mathcal{L}_{\mathrm{adv}}$:** The adversarial losses are applied to the predicted hand and face parameters for in-the-wild data to constrain their parameter space, and for Decaf data to facilitate the training of the discriminators. The adversarial loss is given by:

$$\mathcal{L}_{\mathrm{adv}}(E) = \mathbb{E}_{\boldsymbol{\theta}_f \sim p_E}\left[\log\left(1 - D_F(E(I))\right)\right] + \mathbb{E}_{\boldsymbol{\theta}_h \sim p_E}\left[\log\left(1 - D_H(E(I))\right)\right]. \tag{1}$$

The losses for the hand and face discriminators are given by:

$$\mathcal{L}_{\mathrm{adv}}(D_F) = -\left(\mathbb{E}_{\boldsymbol{\theta}_f \sim p_E}\left[\log\left(1 - D_F(E(I))\right)\right] + \mathbb{E}_{\boldsymbol{\theta}_f \sim p_{\mathrm{data}}}\left[\log\left(D_F(\boldsymbol{\theta}_f)\right)\right]\right), \tag{2}$$

and

$$\mathcal{L}_{\mathrm{adv}}(D_H) = -\left(\mathbb{E}_{\boldsymbol{\theta}_H \sim p_E}\left[\log\left(1 - D_H(E(I))\right)\right] + \mathbb{E}_{\boldsymbol{\theta}_H \sim p_{\mathrm{data}}}\left[\log\left(D_H(\boldsymbol{\theta}_h)\right)\right]\right), \tag{3}$$

where $E$ jointly denotes the image backbone, the mesh encoder and the parameter regressor, $p_E$ denotes the output distribution of $E$, $p_{\mathrm{data}}$ denotes the data distribution of the motion datasets, $\boldsymbol{\theta}_f = (\boldsymbol{\theta}_{\text{f-pose}}, \boldsymbol{\beta}_f, \boldsymbol{\theta}_{\text{f-exp}})$, $\boldsymbol{\theta}_H = (\boldsymbol{\theta}_h, \boldsymbol{\beta}_h)$.

**Depth loss $\mathcal{L}_{\mathrm{depth}}$:** To provide pseudo-3D hand and face keypoints supervision for in-the-wild data, we use a modified SILog Loss (Eigen et al., 2014), an affine-invariant depth loss as our depth supervision $\mathcal{L}_{\mathrm{depth}}$. Formally, let $\hat{K}_D$ denote the pseudo-ground-truth affine-invariant depth of the face and hand keypoints, and $K_D$ denote the rendered depth for the keypoints,

$$\mathcal{L}_{\mathrm{depth}} = \left[\mathbf{Var}\left(\log(K_D + \varepsilon) - \log(\hat{K}_D + \varepsilon)\right)\right]^{1/2}, \tag{4}$$

where $\mathbf{Var}$ is the standard variance operator and $\varepsilon = 10^{-7}$.

Overall, our loss for the mesh and interaction networks is formulated by

$$\mathcal{L} = \lambda_{\mathrm{mesh}}\mathcal{L}_{\mathrm{mesh}} + \lambda_{\mathrm{interaction}}\mathcal{L}_{\mathrm{interaction}} + \lambda_{\mathrm{adv}}\mathcal{L}_{\mathrm{adv}} + \lambda_{\mathrm{depth}}\mathcal{L}_{\mathrm{depth}}, \tag{5}$$

where $\lambda_{\mathrm{mesh}} = 12.5$, $\lambda_{\mathrm{interaction}} = 5$, $\lambda_{\mathrm{depth}} = 2.5$, $\lambda_{\mathrm{adv}} = 1$ for all the experiments in the paper; See more details in Appendix C.

## 4 EXPERIMENTAL RESULTS

### 4.1 DATASETS AND METRICS

**Datasets.** We employ Decaf (Shimada et al., 2023) for reconstructing 3D face and hand interactions with deformations, along with the in-the-wild dataset we collected containing 500 images. We use the shape, pose, and expression data of hands and faces from Decaf (Shimada et al., 2023), RenderMe-360 (Pan et al., 2023a), and FreiHand (Zimmermann et al., 2019) for training the adversarial priors. We use the training set of the aforementioned datasets for network training. We use the official split from Decaf to separate the training and testing sets, and select a few in-the-wild images for the test set to perform qualitative visualizations.

**Metrics.** We adopt commonly-used metrics for mesh recovery accuracy following Kanazawa et al. (2018); Lin et al. (2021a); Dou et al. (2023b); Cho et al. (2022): **1)** *Mean Per-Joint Position Error* (MPJPE): the average Euclidean distance between predicted keypoints and ground-truth keypoints. **2)** *PAMPJPE*: MPJPE after Procrustes Analysis (PA) alignment. **3)** *Per Vertex Error*: per vertex error (PVE) with translation. Following Decaf (Shimada et al., 2023), we use the following metrics to measure the plausibility: **4)** *Collision Distance* (Col. Dist.): the average collision distances over

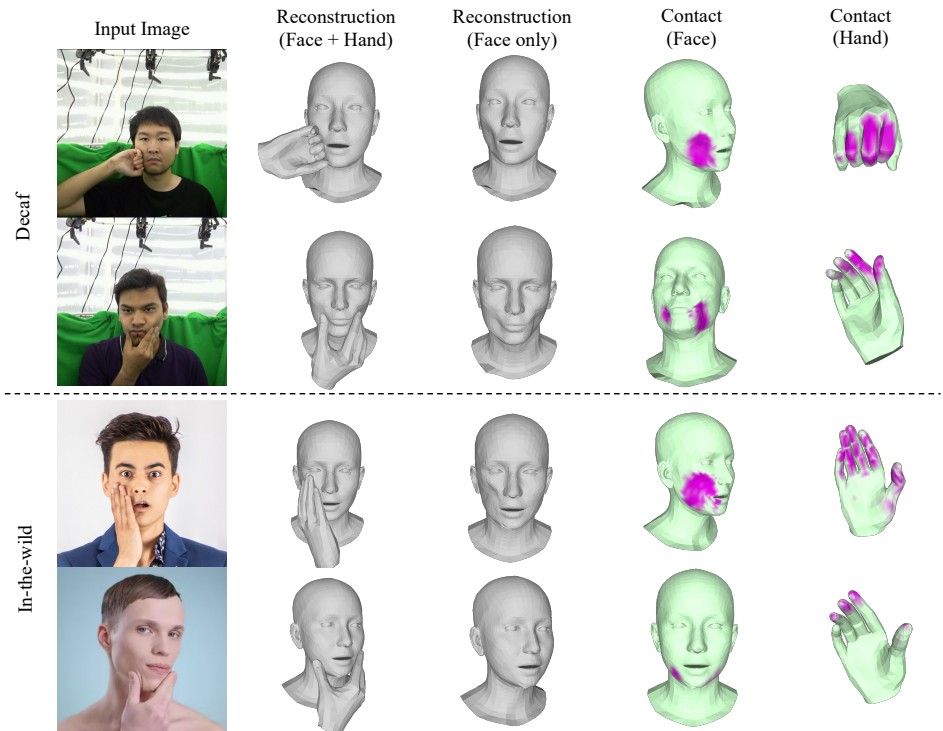

Figure 3: Qualitative results of hand-face interaction, deformation, and contact recovery by DICE on Decaf and in-the-wild images. In contact visualizations, a deeper color indicates a higher contact probability.

vertices and frames; **5**) *Non-Collision Ratio* (Non. Col.): the proportion of frames without hand-face collisions; **6**) *Touchness Ratio* (Touchness): the ratio of hand-face contacts among ground truth contacting frames; **7**) *F-Score*: the harmonic mean of *Non-Collision Ratio* and *Touchness Ratio*. Note that F-Score measures Touchness and Non-Collision Ratio as a whole, which is a metric of overall physical plausibility, whereas Non-Collision Ratio or Touchness are meaningless when considered individually.

## 4.2 IMPLEMENTATION DETAILS

We train MeshNet, InteractionNet, and IKNet, along with the face and hand discriminators using AdamW (Loshchilov, 2017) optimizers, each with a learning rate of $6 \times 10^{-4}$, and a learning rate decay of $1 \times 10^{-4}$. The generator and discriminator networks are optimized in an alternating manner. Our batch size is set to 16 during the training stage. The training takes 40 epochs, totalling 48 hours. The model is trained and evaluated on 8 Nvidia A6000 GPUs with an AMD 128-core CPU. Inference times are calculated on a single Nvidia A6000 GPU.

## 4.3 PERFORMANCE ON HAND-FACE INTERACTION AND DEFORMATION RECOVERY

We compare our method with the following: **1**) **Benchmark**: the baseline (Lugaresi et al., 2019; Li et al., 2017) introduced in Decaf (Shimada et al., 2023); **2**) **Decaf** (Shimada et al., 2023): an optimization-based method for hand-face interaction and deformation recovery. **3**) **PIXIE (whole-body)** (Feng et al., 2021a): a representative model for full-body recovery, including the hand and face, introduced in Decaf. **4**) **PIXIE (hand+face)** (Feng et al., 2021a): a optimization-based variant of PIXIE, introduced in Decaf. For regression-based methods, as we are dealing with a relatively new task, there are few readily available baselines. To facilitate comparison, we adapt the following regression-based models from related tasks: **5**) **METRO** (Lin et al., 2021a): A representative work in human body/hand mesh recovery. We adapt METRO to predict both hand and face meshes, adding extra output heads to predict contact and deformation. **6**) **PIXIE-R** (Feng et al., 2021a): Adapted PIXIE, using the same backbone and hand and face branches but trained with losses from DICE. **7**) **FastMETRO (single-target)** (Cho et al., 2022): Another representative work in human and hand mesh recovery. We adapt two independent FastMETROs, one for estimating hand mesh vertices and contact, and the other for estimating face mesh, deformation, and contact. Here, the word single-target means each FastMETRO considers hand and face individually, with no information exchange. This model is trained using the same hyperparameter, loss, and optimizer as DICE, on the Decaf (Shimada et al., 2023) dataset.

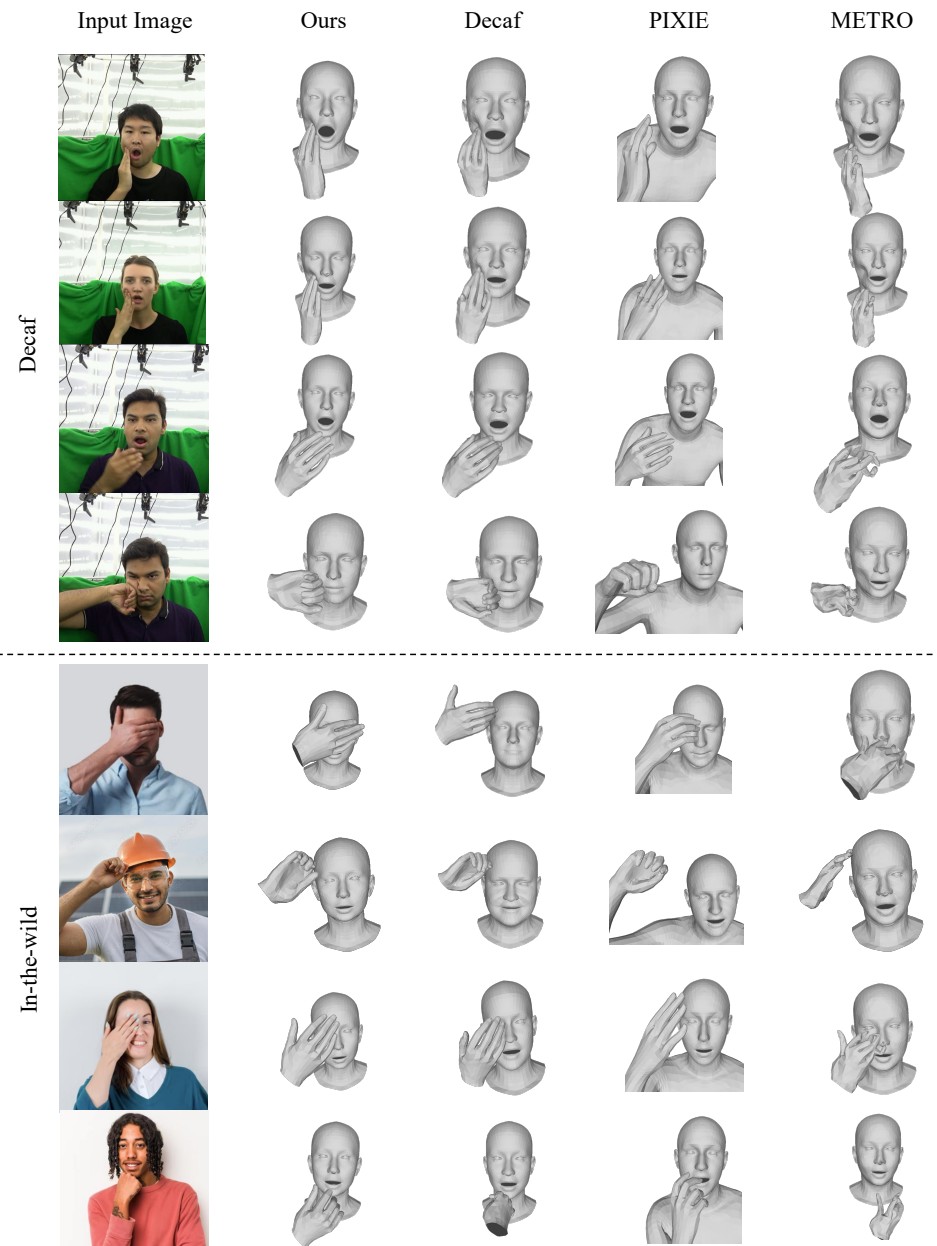

Figure 4: Qualitative comparison of DICE, Decaf (Shimada et al., 2023), PIXIE (Feng et al., 2021a) (whole-body version), METRO* (Lin et al., 2021b) on Decaf validation set and in-the-wild images. Our method achieves superior reconstruction accuracy and plausibility in the Decaf (Shimada et al., 2023) dataset, especially generalizing well to difficult in-the-wild actions unseen in Decaf compared to all baselines.

### 4.3.1 QUANTITATIVE EVALUATIONS

**Reconstruction Accuracy.** In Tab. 1, our method surpasses all baseline methods in terms of reconstruction accuracy, achieving a 7.5% reduction in per-vertex error compared to the current state-of-the-art, Decaf. Note that our method is regression-based and allows inference at an interactive rate, while Decaf (Shimada et al., 2023) uses a cumbersome test-time optimization process, taking more than 200x more time per image. Decaf also requires using temporal information in successive frames, while our method only uses a single frame. Our method shows a 30% reduction in reconstruction error compared to the modified METRO baseline, and up to 79% reduction compared to other end-to-end baselines. Notably, our method achieves a 27% MPVE reduction compared to the PIXIE-R baseline which uses the same mesh and interaction losses as our method, demonstrating the superiority of our network design and weak-supervised training scheme. Our method is also more accurate than another end-to-end baseline, FastMETRO.

Table 1: Comparison of hand-face interaction and deformation recovery on Decaf.

| Methods | Type | 3D Reconstruction Error | | | Physics Plausibility Metrics | | | | Running Time |
|---|---|---|---|---|---|---|---|---|---|
| | | PVE‡↓ | MPJPE↓ | PAMPJPE↓ | Col. Dist. ↓ | Non. Col. ↑ | Touchness ↑ | **F-Score** ↑ | (per image; s)↓ |
| **Comparison between DICE and optimization-based methods** | | | | | | | | | |
| Decaf (Shimada et al., 2023) | O | 9.65 | – | – | 1.03 | 83.6 | 96.6 | **89.6** | 19.59 |
| Benchmark (Lugaresi et al., 2019; Li et al., 2017) | O | 17.7 | – | – | 19.3 | 64.2 | 73.2 | 68.4 | 16.40 |
| PIXIE (hand+face) (Feng et al., 2021a) | O | 26.3 | – | – | 7.04 | 75.9 | 75.1 | 75.5 | – |
| DICE (Ours) | R | **8.32** | 9.95 | 7.27 | **0.16** | 66.6 | 79.9 | 72.7 | **0.088** |
| **Comparison between DICE and regression-based methods** | | | | | | | | | |
| PIXIE (whole-body) (Feng et al., 2021a) | R | 39.7 | – | – | 0.11 | 97.1 | 51.8 | 67.6 | **0.070** |
| PIXIE-R (Feng et al., 2021a) | R | 11.0 | 22.0 | 21.2 | 0.27 | 62.6 | 83.0 | 72.0 | **0.070** |
| METRO* (hand+face) (Lin et al., 2021a) | R | 11.8 | 15.4 | 11.9 | **0.08** | 80.7 | 54.8 | 65.2 | 0.103 |
| FastMETRO* (single-target) (Cho et al., 2022) | R | 9.27 | 11.8 | 9.41 | 0.09 | 82.2 | 55.5 | 66.2 | 0.110 |
| DICE (Ours) | R | **8.32** | 9.95 | 7.27 | 0.16 | 66.6 | 79.9 | **72.7** | 0.088 |

* parametric version. O and R denote optimization-based and regression-based methods, respectively. ‡ calculated after translating the center of the head to the origin. **bold** denotes the best result in a comparison group. Note our method operates at an interactive rate (20 fps; 0.049s per image) on an Nvidia 4090 GPU. Here we report the runtime performance on an A6000 GPU for a fair comparison.

Table 2: Comparison of hand-face contact estimation on Decaf.

| Method | **F-score** ↑ | Precision ↑ | Recall ↑ | Accuracy↑ |
|---|---|---|---|---|
| Decaf (face) (Shimada et al., 2023) | 0.57 | 0.69 | 0.49 | 0.99 |
| Decaf (hand) (Shimada et al., 2023) | 0.47 | 0.62 | 0.39 | 0.98 |
| DICE (face) | **0.61** | 0.64 | 0.57 | 1.00 |
| DICE (hand) | **0.50** | 0.55 | 0.45 | 0.98 |

**Plausibility.** In terms of overall physical plausibility (F-Score), our method is the best among all regression-based methods: PIXIE (whole-body), PIXIE-R, METRO, and FastMETRO. On the other hand, while some optimization-based methods (Decaf and PIXIE (hand+face) have higher overall plausibility (F-Score) compared to DICE, this is due to their test-time optimization, which iteratively adjusts the relative positioning of hand and face. Thus, they are much more computationally intensive than our regression-based method. With a highly efficient end-to-end inference scheme, DICE still outperformed an optimization-based method (Benchmark) on F-Score.

**Contact Estimation.** The contact estimation metrics (accuracy, precision, recall) are calculated by the predicted per-vertex contact probabilities against the respective 0-1 contact ground truths. In Tab. 2, DICE achieves superior contact estimation performance on the Decaf dataset, surpassing previous work (Shimada et al., 2023) in F-Score for both face and hand contacts. Here F-score provides a comprehensive measure of both precision and recall ratio combined. These two metrics involve a trade-off: focusing solely on precision may lead to a decrease in recall, and vice versa. Balancing this trade-off, the F-score offers a more meaningful evaluation of contact estimation.

### 4.3.2 QUALITATIVE EVALUATIONS

As discussed in Sec. 3.2, the Decaf (Shimada et al., 2023) dataset is collected in an indoor environment with a green screen, which doesn't reflect the complex environment where real-world hand-face interactions occur. Therefore, a model only trained with the Decaf dataset might have generalization issues when tested on in-the-wild data. Fig. 4 supports this claim by demonstrating our model's superior generalization performance on in-the-wild data with unseen identity and pose. On the other hand, Decaf's reconstruction suffers from self-collision and incorrect hand-face relationship. PIXIE and METRO reconstruct inaccurate hand poses and often demonstrates implausible non-touching artefacts. As shown in Fig. 3, our method faithfully reconstructs hand-face interaction and deformation and accurately labels the contact areas.

### 4.4 ABLATION STUDY

**Network Design.** In Tab. 3, adopting the two-branch architecture, which separates deformation and interaction estimation from mesh vertices regression, improves both accuracy and plausibility.

**In-the-wild data.** As shown in Tab. 3, adding weak-supervision training and in-the-wild data for DICE training improves all reconstruction error metrics (PVE*, MPJPE, PAMPJPE) while maintaining a high plausibility (F-Score). We deem that the slight decrease in F-Score could mainly be attributed to the difference in distribution between the studio-collected Decaf (Shimada et al., 2023) and in-the-wild data. This is because the limited pose and identity distribution of the Decaf training dataset may cause the model to overfit, and the inclusion of in-the-wild images out of the Decaf data distribution effectively improves the generalization capability of DICE.

Table 3: Comparison of hand-face interaction and deformation recovery on Decaf. **Bold** denotes the best result.

| Methods | PVE*↓ | MPJPE↓ | PAMPJPE↓ | F-Score ↑ |
|---|---|---|---|---|
| DICE (single branch) | 9.29 | 11.6 | 8.51 | 69.3 |
| DICE (w.o. in-the-wild data) | 8.93 | 11.0 | 7.50 | **73.3** |
| DICE (w.o. supervision on $\mathbf{V}'_F, \mathbf{V}'_H$) | 12.2 | 14.4 | 11.1 | 70.7 |
| DICE (w.o. $\mathcal{L}_{\text{depth}}$) | 15.6 | 19.5 | 13.7 | 64.2 |
| DICE (w.o. $\mathcal{L}_{\text{params}}$) | 10.3 | 12.8 | 10.4 | 64.7 |
| DICE (w.o. $\mathcal{L}_{\text{adv}}$) | 11.1 | 14.2 | 10.4 | 69.8 |
| DICE (Full) | **8.32** | **9.95** | **7.27** | 72.7 |

**Unrefined Features Supervision.** Regressing the unrefined head and hand mesh features $\mathbf{V}'_F, \mathbf{V}'_H$ and then perform inverse kinematics to regress the parametric mesh improves plausibility and accuracy, compared to directly estimating the face and hand parameters.

**Depth Supervision.** Although depth supervision is only applied to in-the-wild data, as shown in Tab. 3, removing it also significantly degrades performance on the Decaf validation set. Without depth loss, wrong predictions in depth are not penalized for in-the-wild data, introducing noise in the training process, and resulting in erroneous predictions on the Decaf dataset. As shown in Appendix Fig. 7, the absence of depth supervision introduces ambiguity in the z-direction, resulting in artifacts such as self-collision.

**Parameter Supervision.** Supervising parameters directly, in addition to the indirect supervision of parameters by the mesh losses, improves both plausibility and accuracy. This is because direct parameter supervision eliminates ambiguity, preventing the network from converging to alternative parameter combinations that produce incorrect meshes that appear geometrically similar, *i.e.*, with small vertex loss, to the target but are incorrect in their underlying structure, such as pose or shape.

**Adversarial Prior.** The adversarial prior incorporates diverse but realistic pose and shape distribution beyond Decaf (Shimada et al., 2023), ensuring the reality of regressed mesh while allowing for generalization. As shown in Tab. 3, introducing adversarial supervision improves the accuracy and physical plausibility.

### 4.5 LIMITATIONS AND FUTURE WORKS

While our method achieves SotA accuracy on the Decaf (Shimada et al., 2023) dataset and generalizes well to unseen scenes and in-the-wild cases, it still encounters failure cases when the hand-pose interactions are extremely challenging and have severe occlusions (see Appendix D.2). Moreover, despite our method effectively recovering hand and face meshes with visually plausible face deformations, there remains room for improvement in deformation accuracy and physical plausibility. Hand deformations could also be considered in future work for more realistic reconstructions. In the future, physics-based simulation (Hu et al., 2018; Li et al., 2020; Hu et al., 2019; Han et al., 2019; Lin et al., 2022; Huang et al., 2024) can be used as a stronger prior, producing more physically accurate estimations. In this paper, although we found using 500 in-the-wild images significantly improves the model's generalization ability, scaling up to a larger amount of in-the-wild data, on the order of millions or billions, would further enhance performance, which we will study in future work.

## 5 CONCLUSION

In this work, we present DICE, the first end-to-end approach for reconstructing 3D hand and face interaction with deformation from monocular images. Our approach features a two-branch transformer structure, MeshNet, and InteractionNet, to model local deform field and global mesh geometry. An inverse-kinematic model, IKNet, is used to output the animatable parametric hand and face meshes. We also proposed a novel weak-supervision training pipeline, using a small amount of in-the-wild images and supervising with a depth prior and an adversarial loss to provide pose priors. Benefitting from our network design and training scheme, DICE demonstrates state-of-the-art accuracy and plausibility, compared with all previous methods. Meanwhile, our method achieves a fast inference speed (20 fps), allowing for more downstream interactive applications. In addition to strong performance on the standard benchmark, DICE also achieves superior generalization performance on in-the-wild data.

ACKNOWLEDGMENTS

This work is partly supported by the Research Grant Council of Hong Kong (Ref: 17210222), the Innovation and Technology Commission of Hong Kong under the ITSP-Platform grants (Ref: ITS/319/21FP, ITS/335/23FP) and the InnoHK initiative (TransGP project). The research work was in part conducted in the JC STEM Lab of Robotics for Soft Materials funded by The Hong Kong Jockey Club Charities Trust.

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

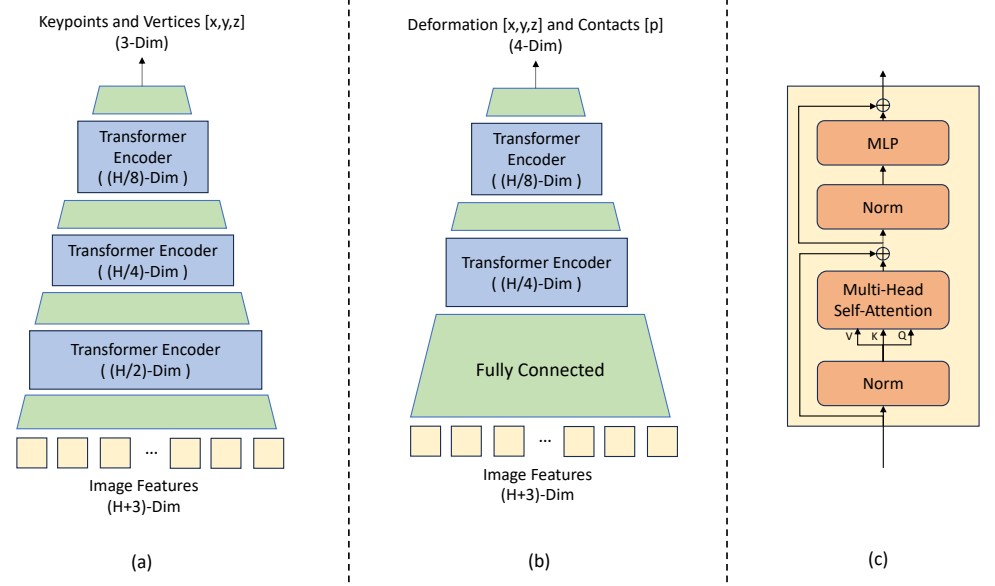

Figure 5: Structural details of the MeshNet and InteractionNet. (a) MeshNet; (b) InteractionNet; (c) Internal structure of a Transformer Encoder block.

# A    IMPLEMENTATION DETAILS

## A.1    CNN BACKBONE

The CNN backbone used in our framework is an HRNet-W64 (Sun et al., 2019), initialized with ImageNet-pretrained weights. The weights of the backbone would be updated during training. We extract a $(49 \times H)$-dim feature map from this network and upsamples it to a $(N \times H)$-dim feature map, where $N = N_{h_k} + N_{f_k} + N_{h_v} + N_{h_v}$, the total number of head and hand keypoints $N_{h_k}, N_{f_k}$ and vertices $N_{h_v}, N_{f_v}$. Then, we concatenate the keypoints and the vertices corresponding to the head and hand mean pose as keypoints and vertex queries, resulting in a $((N + 3) \times H)$-dim feature map. Random masking of keypoints and vertex queries of rate $30\%$ is applied, following (Lin et al., 2021a).

## A.2    MESHNET AND INTERACTIONNET

Our MeshNet and InteractionNet have similar progressive downsampling transformer encoder structures, see Fig. 5 for an illustration. The MeshNet has three component transformer encoders with decreasing feature dimensions. The InteractionNet starts with a fully connected layer that downsamples the feature dimension, followed by two transformer encoders. Each transformer encoder has a Multi-Head Attention module consisting of 4 layers and 4 attention heads. In addition to head and hand mesh features, MeshNet also regresses head and hand keypoints, which are only for supervision and not used by any downstream components.

## A.3    IKNET

Our IKNets take in rough mesh features $\mathbf{V}'_F, \mathbf{V}'_H$ and output the pose and shape parameters $(\theta, \beta)$, as well as the global rotation and translation $(R, T)$. They feature a Multi-Layer Perceptron (MLP) structure, each consisting of five MLP Blocks and a final fully connected layer. Each MLP Block contains a fully connected layer, followed by a batch normalization layer (Ioffe & Szegedy, 2015) and a ReLU activation layer. There are two skip-connections, connecting the output of the first block with the input of the third block, and the output of the third block with the input of the final fully connected layer. See Fig. 6 for an illustration. The hand and head IKNets have the same structure, differing only in their input and output dimensions. The hidden dimensions of the two IKNets are 1024.

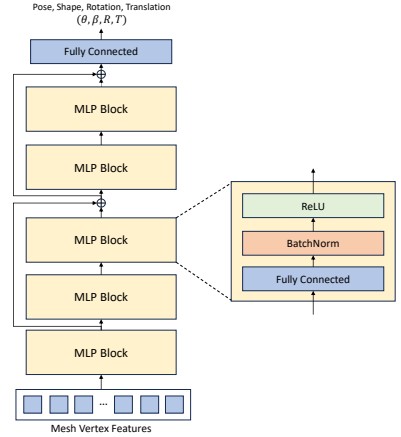

Figure 6: Structural details of the IKNet.

| Input | Without Depth | With Depth |
|---|---|---|

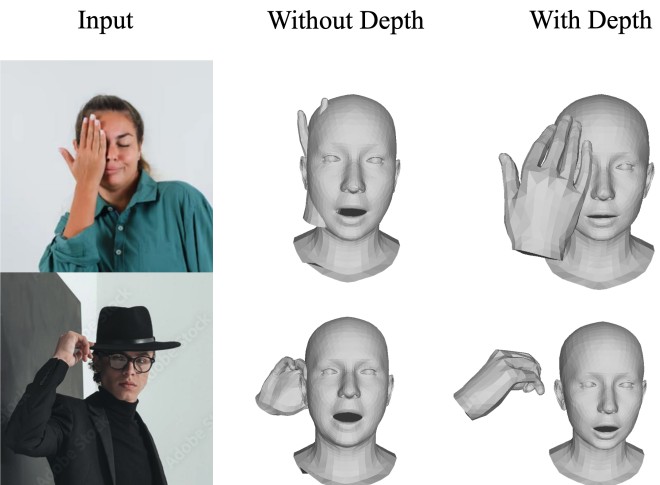

Figure 7: Qualitative demonstration of the effects of the depth loss. The model generalizes poorly in the z-direction when trained without depth supervision.

### A.4 TRAINING AND TESTING DETAILS

To be consistent with the training setting of Decaf[1] (Shimada et al., 2023), in the Decaf dataset, we use all eight camera views and the subjects S2, S4, S5, S7, and S8 in the training data split for training. For testing, we use only the front view (view 108) and the subjects S1, S3, and S6 in the testing data split. The low, mid, and high-resolution head mesh consists of 559, 1675, and 5023 vertices, respectively. The low and high-resolution hand mesh consists of 195 and 778 vertices, respectively. We use the middle-resolution head mesh and the high-resolution hand mesh as the inputs of head and hand IKNets.

## B MORE QUALITATIVE COMPARISONS

We demonstrate qualitatively the effect of the absence of the depth loss in Fig. 7. When trained without depth loss, the network is only supervised with 2D information on in-the-wild data, without any constraints in the z-direction. As a result, artifacts such as self-penetration frequently occur in this case. The introduction of depth loss eliminates this ambiguity, allowing the correct relative positioning of hand and face.

---

[1]Confirmed by the authors of Decaf

## C  ADDITION DETAILS ON LOSSES

Here, we provide the details of the mesh losses and the interaction losses. The details of the adversarial loss and the depth loss are already mentioned in the main paper.

### C.1  MESH LOSSES

The mesh loss $\mathcal{L}_{\mathrm{mesh}}$ consists of four components.

$$\mathcal{L}_{\mathrm{mesh}} = \mathcal{L}_{\mathrm{reproj}} + 4\mathcal{L}_{\mathrm{vert}} + 2\mathcal{L}_{\mathrm{key}} + 2\mathcal{L}_{\mathrm{params}} \tag{6}$$

**Vertices Loss.** $L_1$ loss is used for predicted rough 3D face and hand vertices $\mathbf{V}'_f$, $\mathbf{V}'_h$, FLAME-regressed undeformed 3D face vertices $\mathbf{V}^*_f$ and MANO-regressed 3D hand vertices $\mathbf{V}_h$ against the ground-truth 3D undeformed face vertices $\hat{\mathbf{V}}_f$ and 3D hand vertices $\hat{\mathbf{V}}_h$.

$$\mathcal{L}_{\mathrm{vert}} = \lambda_h(\mu_{\mathrm{nonpara}}\|\mathbf{V}'_h - \hat{\mathbf{V}}_h\|_1 + \|\mathbf{V}_h - \hat{\mathbf{V}}_h\|_1) + \lambda_f(\mu_{\mathrm{nonpara}}\|\mathbf{V}'_f - \hat{\mathbf{V}}_h\|_1 + \|\mathbf{V}^*_f - \hat{\mathbf{V}}_f\|_1) \tag{7}$$

where $\lambda_h, \lambda_f$ are empirically set to 3 and 1 respectively. $\mu_{\mathrm{nonpara}}$ is set to 4 to emphasize the supervision on the more complex non-parametric mesh features.

**Keypoints Loss.** We use $L_1$ loss for predicted rough 3D face and hand keypoints $\mathbf{K}'_f$, $\mathbf{K}'_h$, 3D face and hand keypoints extracted from rough mesh $\mathbf{K}_{f_{\mathrm{mesh}}}, \mathbf{K}_{h_{\mathrm{mesh}}}$, FLAME-regressed 3D face keypoints $\mathbf{K}_f$ and MANO-regressed 3D hand keypoints $\mathbf{K}_h$ against the ground-truth 3D undeformed face keypoints $\hat{\mathbf{K}}_f$ and 3D hand keypoints $\hat{\mathbf{K}}_f$.

$$\mathcal{L}_{\mathrm{key}} = \mu_{\mathrm{nonpara}}(\|\mathbf{K}'_h - \hat{\mathbf{K}}_h\|_1 + \|\mathbf{K}_{h_{\mathrm{mesh}}} - \hat{\mathbf{K}}_h\|_1 + \|\mathbf{K}'_f - \hat{\mathbf{K}}_f\|_1 + \|\mathbf{K}_{f_{\mathrm{mesh}}} - \hat{\mathbf{K}}_f\|_1) \tag{8}$$

$$+ \|\mathbf{K}_f - \hat{\mathbf{K}}_f\|_1 + \|\mathbf{K}_h - \hat{\mathbf{K}}_h\|_1 \tag{9}$$

Where $\mu_{\mathrm{nonpara}}$ is empirically set to 4, to put more weight on the non-parametric mesh with high degrees of freedom.

**Reprojection loss.** $L_1$ loss is used for reprojected rough 3D face and hand keypoints $\mathbf{K}'_f$, $\mathbf{K}'_h$, 3D face and hand keypoints extracted from rough mesh $\mathbf{K}_{f_{\mathrm{mesh}}}, \mathbf{K}_{h_{\mathrm{mesh}}}$, FLAME-regressed 3D face keypoints $\hat{\mathbf{K}}_f$ and MANO-regressed 3D hand keypoints $\hat{\mathbf{K}}_h$ against the ground-truth face and hand 2D keypoints $\hat{\mathbf{K}}_{f_{2\mathrm{D}}}, \hat{\mathbf{K}}_{h_{2\mathrm{D}}}$.

$$\mathcal{L}_{\mathrm{reproj}} = \lambda_h(\|\Pi(\mathbf{K}'_h) - \hat{\mathbf{K}}_{h_{2\mathrm{D}}}\|_1 + \|\Pi(\mathbf{K}_{h_{\mathrm{mesh}}}) - \hat{\mathbf{K}}_{h_{2\mathrm{D}}}\|_1 + \|\Pi(\mathbf{K}_h) - \hat{\mathbf{K}}_{h_{2\mathrm{D}}}\|_1) \tag{10}$$

$$+ \lambda_f(\|\Pi(\mathbf{K}'_f) - \hat{\mathbf{K}}_{f_{2\mathrm{D}}}\|_1 + \|\Pi(\mathbf{K}_{f_{\mathrm{mesh}}}) - \hat{\mathbf{K}}_{f_{2\mathrm{D}}}\|_1 + \|\Pi(\mathbf{K}_f) - \hat{\mathbf{K}}_{f_{2\mathrm{D}}}\|_1) \tag{11}$$

Where $\Pi$ is the learned camera projection function. $\lambda_h, \lambda_f$ are set to 4 and 1 respectively.

**Parameter loss.** We apply $L_1$ loss on the regressed hand and face pose, shape, and facial expression parameters against their respective ground truths.

$$\mathcal{L}_{\mathrm{face\text{-}params}} = (\|\beta_{\mathrm{f}} - \hat{\beta}_{\mathrm{f}}\|_1 + \|\theta_{\mathrm{f\text{-}exp}} - \hat{\theta}_{\mathrm{f\text{-}exp}}\|_1 + \|\theta_{\mathrm{f\text{-}pose}} - \hat{\theta}_{\mathrm{f\text{-}pose}}\|_1)/3 \tag{12}$$

$$\mathcal{L}_{\mathrm{hand\text{-}params}} = (\|\beta_{\mathrm{h}} - \hat{\beta}_{\mathrm{h}}\|_1 + \|\theta_{\mathrm{h}} - \hat{\theta}_{\mathrm{h}}\|_1)/2 \tag{13}$$

$$\mathcal{L}_{\mathrm{params}} = \mathcal{L}_{\mathrm{face\text{-}params}} + \mathcal{L}_{\mathrm{hand\text{-}params}} \tag{14}$$

### C.2  INTERACTION LOSSES.

The interaction loss $\mathcal{L}_{\mathrm{interaction}}$ consists of four components.

$$\mathcal{L}_{\mathrm{interaction}} = 0.2\mathcal{L}_{\mathrm{touch}} + 0.6\mathcal{L}_{\mathrm{contact}} + \mathcal{L}_{\mathrm{collision}} + 6\mathcal{L}_{\mathrm{deform}} \tag{15}$$

**Deformation loss.** Due to the human anatomy, some vertices on the face are more easily deformed than other vertices. Therefore, we impose an adaptive weighting on each vertex, and use square

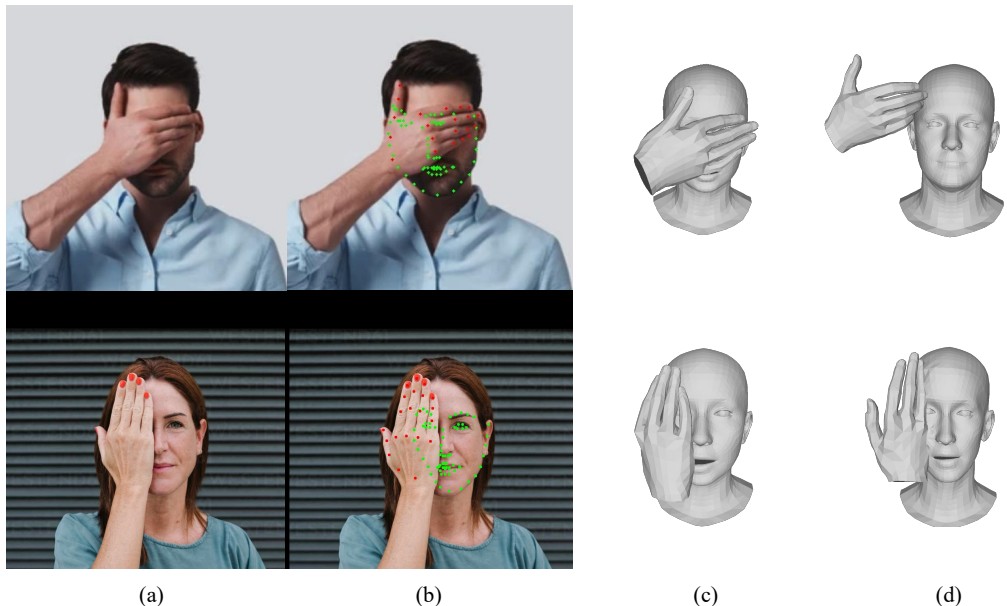

|  (a)  |  (b)  |  (c)  |  (d)  |

Figure 8: Examples of failed keypoint estimation in case of large self-occlusion. (a) input image; (b) inaccurate keypoint estimation by the same keypoint estimators used in Decaf (Lugaresi et al., 2019; Bulat & Tzimiropoulos, 2017); (c) reconstructed hand-face interaction by our method. (d) reconstructed hand-face interaction by Decaf.

loss to penalize large deformation. We also have a regularization term to penalize extremely large deformations.

$$\mathcal{L}_{\text{deform}} = \sum_{i \in \mathcal{I}} (1 + \mu \|\hat{d}_i\|_2) \|\hat{d}_i - d_i\|_2^2 + \lambda \sum_{i \in \mathcal{L}} \|d_i\| \tag{16}$$

Where $\mathcal{I}$ is the set of indices of face vertices, $d_i$, $\hat{d}_i$ are the predicted and ground truth deformation vector for index $i$, and $\mathcal{L} = \{i \in \mathcal{I} : \|d_i\|_2 > 3cm\}$ the vertices of large deformations. $\mu$ and $\lambda$ are empirically set to be 5000, 100 respectively.

**Touch loss.** Let $\mathbf{V}_{F_C}$ and $\mathbf{V}_{H_C}$ denote the set of face and hand vertices that are predicted by the model to have contact probability greater than $0.5$.

$$\mathcal{L}_{\text{touch}} = \text{CD}(\mathbf{V}_{F_C}, \mathbf{V}_{H_C}) + \text{CD}(\mathbf{V}_{H_C}, \mathbf{V}_{F_C}) \tag{17}$$

Where $\text{CD}(X, Y)$ gives the mean Chamfer Distance (CD) between each point in $X$ to the closest point in $Y$.

**Collision loss.** Let $\mathbf{V}_{H_{\text{Col}}}$ denote the set of hand vertices that penetrates the face surface, $\mathbf{V}_F$ and $\mathbf{D}_F$ denote the predicted face mesh vertices and deformations.

$$\mathcal{L}_{\text{collision}} = \text{CD}(\mathbf{V}_{H_{\text{Col}}}, \mathbf{V}_F - \mathbf{D}_F) \tag{18}$$

**Contact loss.** Let $\mathbf{C}_H$ and $\mathbf{C}_F$ denote the predicted hand and face contact probabilities and $\hat{\mathbf{C}}_H$, $\hat{\mathbf{C}}_F$ denote the ground-truth contact labels.

$$\mathcal{L}_{\text{contact}} = \text{BCE}(\mathbf{C}_H, \hat{\mathbf{C}}_H) + \text{BCE}(\mathbf{C}_F, \hat{\mathbf{C}}_F) \tag{19}$$

Where BCE denote the binary cross-entropy loss.

## D MORE DISCUSSIONS

### D.1 PERFORMANCE UNDER CHALLENGING OCCLUSION.

As seen in Fig. 8, our end-to-end DICE method is robust under challenging self-occlusion cases, such as the hand covering more than half of the face. On the other hand, Decaf (Shimada et al., 2023), which requires an initial keypoint prediction for test-time optimization, performs poorly in this situation.

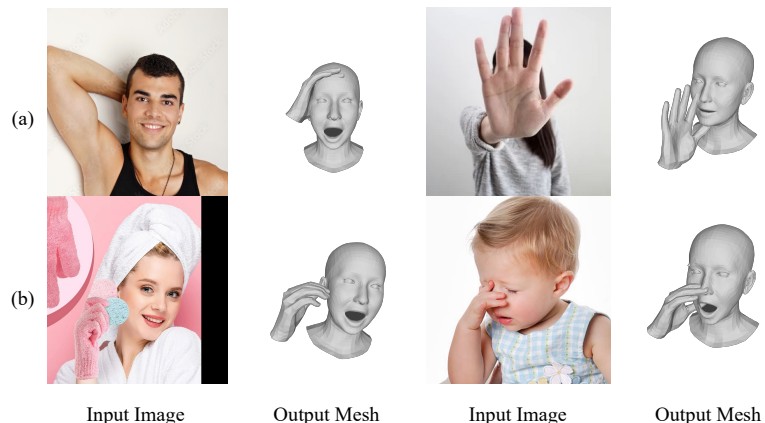

Figure 9: Examples of failure cases in case of complete occlusion of the hand.
(a) Hand or face completely occluded. (b) Out-of-distribution data.

## D.2    FAILURE CASES

In Fig. 9, we demonstrate the failure cases of our method. When the hand is extremely far from the face, or when the hand is completely obscured by the head, our method could fail to reconstruct the hand-face interaction. Also, when given out-of-distribution data, such as when the hand is wearing gloves or the input subject is an infant, the reconstruction accuracy could degrade.

## D.3    SOCIETAL IMPACT

### D.3.1    POTENTIAL MISUSE

DICE enables tracking of individuals' appearances, gestures, and interactions with high fidelity, there is a risk that it may be misused for negative applications, such as surveillance, and may cause privacy infringement. Also, since DICE makes use of a readily animatable representation, it could enable realistic deepfakes driven by the pose and shape information collected, which could be used in creating misinformation and conducting identity theft. We are firmly against any form of misuse of the DICE model.

### D.3.2    DATA FAIRNESS

As hand-face interaction recovery is a human-related task, data fairness is critical. The currently used Decaf Shimada et al. (2023) dataset needs improvement in the inclusion of human actors from underrepresented demographic groups. This may result in a model trained only on Decaf underperforming on input data on such groups, perpetuating inequality and limiting equitable access. Our weak-supervised training scheme introduces diverse in-the-wild data, which could alleviate this issue as the amount of in-the-wild data scales up.

