# OpenReview forum: "DICE: End-to-end Deformation Capture of Hand-Face Interactions from a Single Image"
_ICLR.cc/2025/Conference — ICLR 2025 Poster_

### Official Review · Reviewer_mMEY · 2024-10-17

**Soundness:** 3
**Presentation:** 3
**Contribution:** 2
**Rating:** 6
**Confidence:** 5

**Summary:**

### Motivation
- Modeling the geometry and interactions of human hands and face from a monocular image is an underexplored tasks in 3DCV, with broad applications (AR/VR, motion capture, etc.).
- To the best of the authors' knowledge (and mine), only one prior method tackle this problem in an exhaustive and end-to-end manner, Decaf [Shimada et al., 2023]. This seminal work is however optimization-based, and therefore too slow for interactive use.

### Contributions
- The authors thus introduce DICE, which performs the above-mentioned task in a forward-inference manner. The method relies on a Transformer-based network, with two branches separately tackling the regression of (a) the hand+face geometry and (b) the face deformation caused by hand contact.
- Furthermore, the authors introduce a weak supervision to expand training to in-the-wild-ish images (2D keypoint annotations are still needed, though not 3D ones). Depth supervision is also enabled on such data by integrating a pre-trained monocular depth estimator to generate pseudo ground-truth. Predicted meshes are projected into depth images using a differentiable renderer for comparison.


### Results
- Experiments on the Decaf benchmark show that DICE outperforms prior art in terms of 3D reconstruction accuracy and provides competitive runtime on par with regression-based baselines (~0.1s, whereas Decaf needs ~15s). Unlike prior art though, the authors did not fully disclose their results on physics-plausibility relating to touchness and collision.
- An ablation study, as well as plenty of convincing qualitative results, are also provided.

### Relevance
- This paper addresses a relevant yet under-explored 3DCV task. The community would benefit from this iteration over the seminal Decaf.

**Strengths:**

_(somewhat ordered from most to least important)_

### S1. Clear Explanations and Decent Reproducibility
- The Methodology is clear, thoroughly explained, and overall well-formalized.
- While the actual implementation has not been provided, an expert in the art should be able to re-implement this work.
- The paper is also well illustrated and easy to follow (pipeline figures + qualitative results).
- Limitations are well-discussed, with failure cases provided (Appendix D).

### S3. Convincing Evaluation
- The evaluation is also clear, comprehensive, and well presented (e.g., highlighting optimization-based vs. regression-based methods in Table 1, providing various qualitative results, etc.).
- The proposed DICE method outperforms the SOTA on 3D reconstruction metrics, at a competitive running time.
- Qualitative results and the ablation study are also convincing. In-the-wild examples help illustrate the generalizability of the method.

### S3. End-to-End Solution Targeting an Under-explored Task
- As mentioned above, this paper is seemingly only the second (after Decaf [Shimada et al., 2023]) to address the joint modeling of hand/face geometry and interaction in a comprehensive, end-to-end manner. As such, it would be valuable to the community.
- While the proposed method mostly relies on existing components (CNN-backbone, Transformer processing, weak supervision using large monocular depth regressor, etc.), it holds value as a well-designed end-to-end system.

**Weaknesses:**

_(somewhat ordered from most to least important)_


### W1. Limited Technical Novelty
- The proposed solution has value as an end-to-end system tackling an under-explored task, but its technical novelty is limited.
- The main/only technical contribution claimed by the authors are their weak depth supervision module, relying on a large/foundation model [L110-112]. There might be some novelty in applying such a model to this specific task (e.g., in terms of processing the depth-map predictions to supervise the facial keypoints), but the underlying idea of leveraging a large monocular depth model for weak supervision is not novel.
- Most of the loss functions appear borrowed from Decaf [Shimada et al., 2023], even though this is not made obvious in the paper (both in Section 3.3. and in Appendix C).



### W2. Lack of Insight w.r.t. Physics Metrics
- Unlike Decaf, the authors here did not report all physics-related metrics (non-collision / touchness accuracy).
- The authors could provide more insight in terms of the challenges faced by regression-based methods for accurate modeling, compared to optimization-based solutions.
- Similarly, Table 2 and corresponding paragraph [L417-421] lack proper insight/description. The superiority of DICE is not as obvious as claimed.

### W3. Somewhat Limited Comparison to Prior Art
- The authors compare their method to 3 other solutions: Decaf [Shimada et al., 2023] (most relevant as it also targets hand/face interaction), PIXIE [Feng et al., 2021a] and METRO [Lin et al., 2021a)] (both originally developed for full-body regression), and a baseline/benchmark solution [Lugaresi et al., 2019; Li et al., 2017)] (composed of two independent hand-only and face-only models). While the literature does indeed lack in prior art focusing on hand/face interaction, more baseline methods could have been considered, e.g., mixing other single-target methods (MinimalHand , RingNet, FastMETRO, etc.). This could have further highlighted the benefits on modeling both body parts jointly.
- To counterbalance this "weakness", it should be noted that the aforementioned set of prior methods have been entirely borrowed from Decaf, so the authors here stuck to Decaf's protocol, which is fair. More comparisons could make the paper stronger.

### W4. Minor Remarks
- Societal impact is not discussed, showing a lack of self-reflection. While positive applications are actually listed in the Introduction (e.g., benefits to AR/VR and gaming industry), a variety of negative applications can also come to mind: surveillance (appearance + gesture), animatable deepfakes, etc. Discussions on data fairness/bias are also always relevant when it comes to human-modeling solutions.

**Questions:**

_see **Weaknesses** for key questions/remarks._


### Q1. Hand Rigidity $\rightarrow$ Applicability of CONTHO
- One limitation of this method, as well as prior Decaf, is to only consider face deformation and ignore hand one. While the hands are definitely _stiffer_, ignoring their soft tissues can yield incorrect results in some cases (e.g., hand pressed against forehead). Moreover, by considering the hand as piece-wise rigid, the target task becomes similar to the much less under-explored domain of human/object interaction modeling (i.e., both tasks can be expressed as modeling 3D non-rigid/rigid interactions). It seems that recent solutions such as CONTHO [Nam et al., 2024] could be adapted to face/hand reconstruction. What is the authors' position w.r.t. these adjacent methods? How would the authors differentiate their own approach?

---

> ### Author Response · Authors · 2024-11-19
>
> Thank you for your detailed and insightful review. We appreciate that you acknowledged our clarity in explanations, comprehensive evaluation, and the targeting of an under-explored task. We address your questions and concerns below.
>
> > Novelty of applying large depth model for weak supervision
>
> Thank you for your comment! We believe that the spatial relationship is critical for mesh recovery and interaction modeling, which is naturally encoded by the depth estimation results. This insight motivates us to use depth prior to better tackle our problem.
>
> Notably, while previous works utilize large depth estimation models, they mainly utilize pixel-to-pixel correspondence to supervise the predicted depth and normal map. Instead, we proposed our novel pipeline to perform depth supervision only on keypoints. By employing lightweight and effective keypoint-to-keypoint correspondence, our method could provide robust depth supervision even when the predicted mesh is not well-aligned with the ground truth on the x-y plane.
>
> Another example of our insight is that in addition to the depth and 2D keypoints supervision, we also employed pose/shape priors from existing motion datasets. In addition to spatial supervision, the pose/shape priors constrain the parameters of the parametric models to conform to human anatomy. We validated this design in Sec. 4.4 and showed that adding pose/shape priors improves accuracy and plausibility.
>
>
> Besides the standard 3D reconstruction losses and the interaction losses from Decaf, we proposed the adversarial losses and depth losses to facilitate the use of pose and depth priors for in-the-wild data supervision.
>
> > Insight in terms of the Physics Metrics
>
> We apologize for omitting the Touchness and Collision Ratio terms. We include them here for your reference:
>
> **Comparison between DICE and Optimization-based (Opti.) methods:**
>
> | Model             | Type  | PVE  | MPJPE | PAMPJPE | Col. Dist | Non. Col. | Touchness | F-Score |
> | ----------------- | ----- | ---- | ----- | ------- | --------- | -------- | --------- | ------- |
> | Decaf             | Opti. | 9.65 |   /   |    /     | 1.03      | 83.6     | 96.6      | **89.6**   |
> | Benchmark         | Opti. | 17.7 |   /    |    /     | 19.3      | 64.2     | 73.2      | 68.4    |
> | PIXIE (hand+face) | Opti. | 26.3 |   /    |    /     | 7.04      | 75.9     | 75.1      | 75.5    |
> | DICE              | Reg.  | **8.32** | **9.95** | **7.27**    | **0.16**     | 66.6     | 79.9      | 72.7    |
>
> **Comparison between DICE and Regression-based (Reg.) methods:**
>
> | Model              | Type  | PVE  | MPJPE | PAMPJPE | Col. Dist | Non. Col. | Touchness | F-Score |
> | ------------------ | ----- | ---- | ----- | ------- | --------- | -------- | --------- | ------- |
> | PIXIE (whole-body) | Reg.  | 39.7 |  /     |    /     | 0.11      | 97.1     | 51.8      | 67.6    |
> | PIXIE-R            | Reg.  | 11.0 | 22.0  | 21.2    | 0.27      | 63.6     | 83.0      | 72.0    |
> | METRO              | Reg.  | 11.8 | 15.4  | 11.9    | **0.08**     | 80.7     | 54.8      | 65.2    |
> | DICE               | Reg.  | **8.32** | **9.95**  | **7.27**    | 0.16      | 66.6     | 79.9      | **72.7**    |
>
> Note that Touchness and Non-Collision Ratio complement each other and should **not** be considered **individually**, while F-Score measures the balance between the two values, which is a better metric for evaluation as also witnessed by its use in detection/classification tasks. We deem the reason for limited performance on physical plausibility as follows. While the regression-based method could accurately predict the pose, shape, and overall positioning, the interaction is much harder to accurately predict in a single forward pass. This is because the threshold for touching is +/- 5mm of the mesh surface, and is only decided by the closest parts between the hands and faces (e.g. the fingertip touching face). Therefore, the interaction metrics are mostly decided by details such as the contacting finger’s orientation and position, which is a much smaller detail compared to the positioning of the hand and face that can be relatively more efficiently learned by the regression model, thus significantly harder to regress accurately in a single forward pass.

---

> ### Author Response · Authors · 2024-11-19
>
> > Table 2 (contact estimation) and corresponding paragraph
>
> Thanks for pointing this out. The contact label estimation task is a binary classification problem for each vertex, with the model predicting a probability for contact (labeled 1) and no contact (labeled 0). The accuracy, precision, and recall are calculated in terms of the per-vertex contact ground-truths (0 or 1) for the predicted contact probability, while the F-Score combines precision and recall, evaluating the classification performance holistically. Note that, precision and recall are complementary in nature, and are meaningless when viewed individually. We have a higher F-Score for both hand and face contact estimation compared to Decaf, establishing our method’s better performance.
>
> We will add more details in terms of contact estimation in the revised paper.
>
> Regarding the comparison of single-target methods, we will conduct experiments on FastMETRO, where hand and face are reconstructed with separate networks. We will share the experiment results once they are ready.
>
>
>
>
>
> > Societal Impact
>
> Thank you for your insightful comments on the social impact issues of DICE. We totally agree that societal impact should be discussed more in our paper, and we include some additional discussion down below.
>
> DICE indeed has the risk of being applied to negative applications, as it enables tracking of individuals' appearances, gestures, and interactions with high fidelity, which could have use cases in surveillance and may cause privacy infringement. Also, since DICE makes use of a readily animatable representation, it could enable realistic deepfakes driven by the pose and shape information collected, which could be used in creating misinformation and conducting identity theft.
>
> As hand-face interaction recovery is a human-related task, data fairness is critical. The currently used Decaf dataset needs improvement in the inclusion of human actors from underrepresented demographic groups. This may result in the current DICE model underperforming on input data on such groups, perpetuating inequality and limiting equitable access.
>
> We will definitely include more discussions on societal impact in our revised version.
>
> Thank you again for your insights and careful evaluation of our paper.
> > Hand rigidity could also be considered using CONTHO (human object interaction)
>
> Thank you for your insights in mentioning this exciting future direction.
>
> As mentioned in our limitation section, we assumed that hands are completely rigid in our work, which could yield incorrect results in some cases, such as when the hand is pressed against the forehead. This is a valuable direction for future works to explore.
>
> Compared to CONTHO, a hand-object interaction method, our task is more challenging. This is because the hand-object interaction task assumes both the hand and the object are rigid, while our method takes into account the additional factor of face deformation, which has a high degree of freedom and adds another dimension of complexity to our task.
>
> The main insight brought by the CONTHO paper is that it utilizes a refinement module, which bridges the gap between the pure single-pass end-to-end method and the optimization-based method. In future work on hand-face interaction, a similar module can also be developed to improve the interaction estimations in our problem.

---

> ### Author Response · Authors · 2024-11-23
> **Additional Experimental Results**
>
> Dear reviewer,
>
> As promised, we have finished the FastMETRO (single-target) experiment and the result is as follows:
>
> | Model                        | PVE         | MPJPE     | PAMPJPE     | ColDist     | Non. Col. | Touchness | F-Score     | Runtime  |
> | ---------------------------- | ----------- | ----------- | ----------- | ----------- | --------- | --------- | ----------- | --------- |
> | FastMETRO (single-target)    | $9.27$        | $11.8$        | $9.41$        | $\mathbf{0.09}$    | $82.2$      | $55.5$      | $66.2$        | $0.110$     |
> | DICE (w.o. in-the-wild data) | $\underline{8.93}$ | $\underline{11.0}$ | $\underline{7.50}$ | $\underline{0.11}$ | $74.6$     | $71.9$      | $\textbf{73.3}$    | $\textbf{0.088}$ |
> | DICE (with in-the-wild data) | $\textbf{8.32}$    | $\textbf{9.95}$    | $\mathbf{7.27}$    | $0.16$        | $66.6$      | $79.9$      | $\underline{72.7}$ | $\textbf{0.088}$ |
>
> *Models are tested on the Decaf validation set. Bold denotes the best result and underline denotes the second best result.*
>
> The FastMETRO implementation we tested consists of two independent FastMETRO-L models, each with an HRNet-W64 backbone: The Face Model, responsible for regressing face mesh parameters, face deformation, and face contact; and the Hand Model, which regresses the hand mesh parameters and hand contacts. The model was trained using the same hyperparameters and optimizer as DICE, on the Decaf dataset for 30 epochs.
>
> Our experiment results suggest that both the full DICE model trained with the weak-supervision pipeline outperform the FastMETRO (single-target) model on both accuracy (PVE, MPJPE, PAMPJPE) and plausibility (F-Score). DICE also has a faster runtime per image compared to the tested FastMETRO (single-target) model. Notably, for a fair comparison, the DICE model trained only on the Decaf dataset (without in-the-wild data) could outperform FastMETRO (single-target) trained with the same data and hyperparameter configuration, highlighting our model's superior architecture design. We will add this discussion in our revision.
>
> We hope these results address your concerns. If you have any further comments, please feel free to share them. We sincerely thank you for your time and effort in helping to improve this work.

---

> > ### Comment · Reviewer_mMEY · 2024-11-26
> >
> > I thank the authors for addressing my remarks. I believe that the provided results and discussion would benefit the submission, and I would suggest to the authors that they update their paper and/or supplementary material accordingly.
> >
> > As summarized by  Reviewer `6ptN`, the main concern w.r.t. this submission is its limited technical novelty. That being said, this work could still benefit the community by tackling an under-studied task in an efficient and thorough manner. Therefore, I am still leaning towards acceptance.

---

> ### Author Response · Authors · 2024-11-27
> **reply to Reviewer mMEY**
>
> Dear Reviewer,
>
> Thank you very much for your thoughtful reminder.  As per your guidance, we have now incorporated all the feedback and additional qualitative/quantitative results into a new version of our paper with changes highlighted in the system.
>
> We would sincerely thank you for your time and efforts in reviewing this submission and helping us improve the quality of this work.
> Thank you!
>
> The Authors.

---

### Official Review · Reviewer_fJBo · 2024-10-31

**Soundness:** 3
**Presentation:** 3
**Contribution:** 3
**Rating:** 6
**Confidence:** 4

**Summary:**

The paper introduces a novel method for reconstructing 3D hand-face interactions with deformations from a single image. The authors present DICE, a Transformer-based architecture that simultaneously estimates hand and face poses, contacts, and deformations. A key feature of DICE is its ability to disentangle local deformation fields from global mesh vertex locations into two separate network branches, which enhances the accuracy of deformation and contact estimation for precise hand-face mesh recovery. Additionally, the method utilizes a weakly-supervised training approach with in-the-wild images that lack 3D ground-truth annotations, thereby improving the model's generalizability. Experiments show state-of-the-art performance in terms of accuracy and physical plausibility, along with interactive inference rates on modern GPUs.

**Strengths:**

* The motivation is compelling. DICE provides the first end-to-end learning-based method for capturing hand-face interactions and deformations from a single image, filling a significant gap in the field.
* Good experimental results. The paper claims that DICE achieves superior reconstruction quality compared to baseline methods while operating at an interactive rate (20 fps), which is crucial for real-time applications.
* Technically sound. The weakly-supervised training scheme using in-the-wild images enhances the model's ability to generalize beyond the constraints of studio-collected data.

**Weaknesses:**

* DICE is better than optimization-based methods like Decaf. However, there is a notable gap in physics plausibility metrics compared to Decaf. Can the authors provide a clearer explanation? I wonder if it struggles to balance reconstruction accuracy and physics plausibility.
* In Table 3, using in-the-wild data negatively impacts the F-score. The authors should provide a clearer explanation.
* The novelty is limited. The techniques used in this paper are commonly employed in the community.
* The authors are encouraged to include more discussion on failure cases.
* Some results are not convincing. In Figure 4, the head reconstruction of Decaf is noticeably better than that of the authors' method for the third sample from in-the-wild data.
* Some videos should be included to demonstrate real-time capability.

**Questions:**

Explain the notable gap in Decaf regarding physics plausibility metrics.

---

> ### Author Response · Authors · 2024-11-19
>
> Thank you very much for your insightful and valuable comments. We are encouraged to hear that you acknowledge our motivation, experiment results, technical soundness, and your recognition that we are the first end-to-end method for the hand-face interaction recovery task. We carefully address your questions and concerns below.
>
> > Comparison of Physical Plausibility with Decaf
>
> Thank you for the comment. Decaf’s performance on physical plausibility comes with its **computationally intensive** optimization that iteratively adjusts the relative positioning of the hand, face, and the deformation field, to achieve less interpenetration and more touching. In contrast, our regression method estimates hand face location, pose, and deformation only in a single forward pass, which is more challenging. We chose to build this data-driven end-to-end pipeline because 1) it offers greater efficiency, enabling fast inference to support time-sensitive downstream applications; 2) such a data-driven pipeline could benefit from additional data priors and easily scale up training data to enhance performance.
>
> > Reconstruction Accuracy and Physical Plausibility
>
> Reconstruction accuracy and physical plausibility do not conflict, as perfect accuracy should naturally result in perfect plausibility. However, in our experiments, we found that a small reconstruction error, which is a small part of the overall accuracy calculation, such as the slight deviation of a contacting finger, could result in a penetration or non-touching scenario. We will include this discussion in our revision. Also, we think this is a good point for enhancing physical plausibility in the future works.
> > Effect of in-the-wild data on F-Score
>
> Thanks for the insightful observation! We deem that the slight decrease in F-Score is mainly because the difference between Decaf and the real-world data distribution. Images of the Decaf training set are limited to five actors and a common indoor green-screen background. Therefore, introducing in-the-wild data, which differs from the Decaf distribution in terms of identity, pose, and background, could change the training distribution, and lead to a slight decrease in performance on the Decaf test set during the evaluation. We hope that a more diverse hand-face interaction dataset could be introduced in the future to provide a fair and comprehensive baseline for evaluating real-world hand-face interaction recovery performance. In our paper, to alleviate this, we investigate the benefit of the introduction of in-the-wild data by providing qualitative examples in Figure 4, showing our superior performance of hand-face interaction reconstruction for in-the-wild images compared to the other baseline models and Decaf.
> > The novelty is limited due to using existing components?
>
> Thank you for your comments! Although the use of some components and techniques in our paper are inspired by previous works, our work cannot be taken as a naive combination of existing techniques.
>
> For example, our decision to use the depth prior from a large foundation model is based on this task’s nature, as the 3D recovery of hand and face meshes and reconstructing spatial interaction both require spatial guidance, where depth signals are highly informative and useful.
>
> Moreover, many previous works that use large depth models utilize pixel-to-pixel correspondence to supervise the predicted depth and normal map. Instead, we proposed our novel pipeline to perform depth supervision only on keypoints. By employing lightweight and effective keypoint-to-keypoint correspondence, our method could provide robust depth supervision even when the predicted mesh is not well-aligned with the ground truth on the x-y plane.
>
> We found that only using depth supervision is insufficient and could lead to degenerated results. Thus, we propose a data-driven prior for regularization in addition to the depth and 2D keypoints supervision. Specifically, we employed pose/shape priors from existing motion datasets to constrain the parameters of the parametric hand/face models and to encourage the reconstructed mesh to conform to human anatomy. We also validated this design in Sec. 4.4 and showed that adding pose/shape priors improves accuracy and plausibility.

---

> ### Author Response · Authors · 2024-11-19
>
> > More Discussion on Failure Cases
>
> Thank you for the suggestion! Please take a look at the samples of our failure cases in Fig. 9 in our Appendix. We will include more quantitative samples and discussions in the same section in the revision.
>
> Our method is robust and accurate in general, as shown by the performance under large occlusions in Fig. 4. Failure cases mainly happen when there is extreme occlusion, such as when the hand or the face is completely occluded. Our method also may fail when the input data is too far away from the training distribution, such as when a hand is wearing opaque red gloves and interacting with the face since our dataset only contains bare hands.
>
> > Fig. 4 Decaf Head Reconstruction
>
> Thank you for your insights and your careful observation. We agree that Decaf sometimes provides accurate head reconstructions, owing to its iterative fitting optimization process. However, overall, our method performs better than Decaf on in-the-wild images, especially for predicting the hand pose and hand-face spatial relationship, as shown in Fig. 4. We will supplement more results in our revision. Moreover, as shown by the experiment results, Decaf is not as accurate as our method in hand-face interaction recovery.
>
> > Videos
>
> Thanks for pointing out. We reported the runtime performance in Table 1 of our paper. We uploaded a supplementary video on Google Drive containing a demo where DICE is run on video input. Please check it out.
>
> The link is: https://drive.google.com/file/d/1mj-nOEoDDnqssaie6bvOfh_K1hezjQma/view?usp=sharing. The runtimes of DICE and the baseline methods are reported at 3’15’’ and the real-world application video demo is at [3’30’’’ - 3’47’’’].

---

> ### Author Response · Authors · 2024-12-02
>
> Dear Reviewer,
>
> Thank you very much for your thorough and insightful review. We deeply appreciate the time and effort you’ve invested. As the discussion period is closing, we wanted to kindly ask if our responses to your questions and concerns have addressed them satisfactorily. If there are any remaining questions or areas where further clarification is needed, we would be happy to provide additional information. Thank you!

---

### Official Review · Reviewer_6ptN · 2024-11-03

**Soundness:** 3
**Presentation:** 3
**Contribution:** 2
**Rating:** 6
**Confidence:** 4

**Summary:**

This paper presents a model for the simultaneous reconstruction of face and hand meshes from a single image, based on previous work and annotated data from Shimada et al. (DECAF). The model employs a Transformer-based image encoder to extract keypoints for face and hand through separate streams, utilizing parametric models for reconstruction. Designed for images that contain both a face and a hand, the approach extends training with unannotated data by leveraging a diffusion-based depth estimation model (Ke et al., 2024) to guide mesh reconstruction. After reconstructing the face and hand mesh, the depth maps are computed and aligned with depth estimates from the diffusion model. The model is trained with four types of losses: mesh loss and contact-point loss, similar to those in DECAF, as well as depth loss and adversarial loss, which support the depth supervision process. Experimental results show improvements over previous work for face-hand interaction images.

**Strengths:**

-- The use of depth supervision for weakly-supervised training in mesh reconstruction is innovative and potentially impactful for similar tasks.

**Weaknesses:**

-- The model is focused specifically on face-hand images, which may limit its relevance to the broader computer vision community. The emphasis on this specific interaction appears to be largely application-driven, leaving questions about its applicability to other computer vision tasks.

-- Although the use of depth supervision in training mesh models appears novel, it leverages known components, which slightly reduces the overall technical novelty.

**Questions:**

Is it possible to apply depth supervision to other reconstruction tasks? What makes it particularly suited to the hand-face problem, aside from data availability? Could this approach be more broadly relevant to visual tasks involving contact points between two objects? I wonder if this could be explored further.

---

> ### Author Response · Authors · 2024-11-19
>
> First, we would like to sincerely thank you for your insightful and valuable feedback. We address your questions and concerns below:
>
> > Applicability to other tasks  (Responding to Weakness, Q1)
>
> Thank you for your comments!
>
> We chose to focus on the hand-face interaction task because, although specific, we think this task is critical. Hand-face interaction is a common behavior observed up to 800 times per day across all ages and genders (Spille et al., 2021). It also has numerous applications in graphics and vision systems like VR and AR, games, HCI, etc.
>
> We believe focusing on hand-face interactions could also inspire other fields such as hand-body interaction and hand-object interaction, as witnessed by many recent works (Decaf (SIGGRAPH Asia 2023), CG-HOI (CVPR 2024), MACS (3DV 2024), MoCapDeform (3DV 2022), HULC (ECCV 2022), TUCH (CVPR 2021), ProciGen (CVPR 2024), VisTracker (CVPR 2023), CHORE (ECCV 2022)).
>
> In general, we believe application-driven research could lead to breakthroughs that later influence general-purpose solutions. For example, research on human pose estimation and human mesh recovery, initially for motion capture, has impacted broader areas like computer animation, activity recognition, and robotics. Similarly, we hope that DICE could contribute to motion capture for human modeling, animation, and Metaverse in the future.
>
> Moreover, hand-face interaction recovery could impact beyond the vision community, as contact behavior analysis plays a pivotal role in public health and human behavior studies. For example, it can facilitate contaminated surface tracking and virus transmission analysis.
>
>
> > Novelty of depth supervision on keypoints (Responding to Weakness, Q2)
>
> Thank you for your kind comments. We are encouraged to hear the acknowledgment of our novelty in leveraging the depth prior of vision foundation models.
>
> Although we are using a depth supervision model, it is **not** a naive application of the existing method. We proposed additional key designs to better incorporate the knowledge of the large depth model in our task, e.g.,  Depth estimation models have been used in tasks like video generation and animation generation, where a dense depth map, is employed for supervision. In contrast, in this paper, we apply depth supervision only on keypoints. This is a non-trivial design choice: Our method enables accurate depth supervision even when the rendered hand/face mesh and the ground-truth meshes are misaligned. This is achieved by establishing *keypoint-to-keypoint* correspondence, an efficient and lightweight representation. Under this representation, each depth value at a keypoint on the rendered depth map is more robustly supervised by its counterpart on the ground-truth depth map by associating a predicted keypoint with the corresponding ground-truth keypoint.
>
> Another key reason our depth-supervision on keypoints approach works is that the task involves regressing **parametric models** for the hand and head. The 3D keypoint information provides critical features and sufficient constraints for the regression of pose and shape parameters of the hands and faces.
>
>
> > Applying depth supervision to other reconstruction tasks
>
> Yes, we agree with you. Although we applied depth prior with some key designs, we believe that the depth supervision pipeline is not limited to human tasks.
> The framework of DICE (data prior from a large foundation model + prior for regularization with end-to-end learning) can be generalized, and we hope it could further benefit other monocular 3D reconstruction tasks that suffer from depth ambiguity, augmenting 2D image signals with essential 3D information like depth, normal signals. Besides, our pipeline could help cost-effectively annotate data, paving the way for scaling the number of data for 3D reconstruction tasks and data-driven animation tasks.
>
> > What makes it particularly suited to the hand-face problem, aside from data availability?
>
> The motivation for using depth signals in this task stems from the belief that spatial information is crucial for accurate face and hand mesh recovery, with depth providing strong guidance. Additionally, the interaction between the hand and face fundamentally relies on spatial information, where depth can play a significant role. Therefore, we incorporate depth as a strong spatial prior to enhance both reconstruction and interaction recovery. Furthermore, our approach is motivated by the desire to enable an end-to-end learning pipeline capable of leveraging in-the-wild, unannotated data.
>
> > Applicability of depth supervision to contact points between two objects
>
> Yes, this is a good point! Our approach could also be helpful for the visual task of two-object contact, especially when the input setting is monocular.
> Since monocular image input results in depth ambiguity, introducing depth supervision could help align the contacting objects in the depth direction and reduce spatial error in reconstruction.

---

> > ### Comment · Reviewer_6ptN · 2024-11-21
> > **Response**
> >
> > I have read the authors' response and the other reviews, and I would like to thank the authors for their detailed clarifications. After reviewing their explanations, I am more convinced of the contribution and significance of this work to the hand-face interaction problem. However, I still hold slight concerns about the level of technical novelty, as the approach primarily builds on existing components. It seems this concern is shared by all three reviewers, indicating a broader consensus on this point. Nevertheless, the work demonstrates clear advancements for this specific task, and I appreciate the effort to address the reviewers' questions. I am willing to align my score with that of the other reviewers.

---

> > > ### Author Response · Authors · 2024-11-21
> > > **Response to Reviewer R6ptN**
> > >
> > > We are truly grateful for your constructive comments and the time you invested in evaluating our work. We sincerely appreciate your thoughtful feedback and your kind acknowledgment of our efforts to address the questions raised during the review process. As promised, we will carefully clarify the points you highlighted and make further improvements to the quality and presentation of our paper in the revision. Thank you!

---

### Author Response · Authors · 2024-11-19

We thank all the reviewers for their valuable feedback. We are encouraged that most of the reviewers have recognized our work’s strengths, including acknowledging the novelty of our weak-supervised training pipeline (Reviewer 6ptN), the introduction of the first end-to-end method (Reviewer fJBo, mMEY), good experimental results (Reviewer fJBo, mMEY), and addressing an underexplored task (Reviewer mMEY).
In each response, we will thoroughly address your concerns, providing detailed explanations to clarify the points and answer your questions.

---

### Meta-Review · Area_Chair_s4MJ · 2024-12-19

**Metareview:**

This work introduces an end-to-end method for capturing hand-face interaction deformations from a single image. Accurately recovering hand-face interactions with plausible deformations is significant due to its broad applications in AR/VR.
The proposed approach includes three key innovations:
	1.	Utilizing an attention mechanism to effectively model the relationship between the hand and face.
	2.	Dividing the regression process into two branches to disentangle global geometry from local interaction.
	3.	Learning an intermediate non-parametric mesh representation instead of directly regressing hand and face parameters.
As a novel task, the method is compared with the state-of-the-art Decaf, demonstrating superior performance and improved inference efficiency.

Three reviewers have provided positive feedback on this work, though concerns were raised about its technical novelty, as it integrates existing techniques. The AC has reviewed the paper and discussion during rebuttal period, acknowledging this work's efforts in addressing hand-face deformations from a single image. The pipeline, composed of multiple modules, is well-documented, with insights into each design clarified by the authors. The AC encourages the release of the codes and datasets used in this work to further contribute to this emerging field.

**Additional Comments On Reviewer Discussion:**

The authors have made significant efforts to address the reviewers’ concerns by conducting additional experiments, incorporating detailed discussions, and providing supplementary materials.
For the specific concerns raised:1.	Technical Novelty: The authors clarified that their method introduces key innovations, such as using keypoint-based depth supervision and leveraging pose/shape priors to improve mesh recovery. While they acknowledged the use of existing components, they emphasized the novel integration tailored for this task. 2.Evaluation Metrics: Missing physics metrics, such as touchness and collision ratio, were included. The authors also explained the challenges their regression-based method faces in capturing fine-grained interaction metrics compared to optimization-based methods like Decaf. 3.Failure Cases: Examples of failure cases, such as those involving extreme occlusions or deviations from the training data, were added to the appendix, along with discussions on potential improvements for robustness.
These revisions contribute meaningfully to addressing the concerns and support a positive evaluation of the work.

---

### Decision · Program_Chairs · 2025-01-22

Accept (Poster)